# Structural basis of early translocation events on the ribosome

Emily J. Rundlet[1,2], Mikael Holm[1], Magdalena Schacherl[3], S. Kundhavai Natchiar[1], Roger B. Altman[1], Christian M. T. Spahn[3], Alexander G. Myasnikov[1] & Scott C. Blanchard[1✉]

Peptide-chain elongation during protein synthesis entails sequential aminoacyl-tRNA selection and translocation reactions that proceed rapidly (2–20 per second) and with a low error rate (around $10^{-3}$ to $10^{-5}$ at each step) over thousands of cycles[1]. The cadence and fidelity of ribosome transit through mRNA templates in discrete codon increments is a paradigm for movement in biological systems that must hold for diverse mRNA and tRNA substrates across domains of life. Here we use single-molecule fluorescence methods to guide the capture of structures of early translocation events on the bacterial ribosome. Our findings reveal that the bacterial GTPase elongation factor G specifically engages spontaneously achieved ribosome conformations while in an active, GTP-bound conformation to unlock and initiate peptidyl-tRNA translocation. These findings suggest that processes intrinsic to the pre-translocation ribosome complex can regulate the rate of protein synthesis, and that energy expenditure is used later in the translocation mechanism than previously proposed.

Faithful translocation requires the ribosome to maintain hold of diverse mRNA and tRNA cargo (the tRNA$_2$–mRNA module) while simultaneously allowing their rapid movement between the large and small ribosomal subunits (LSU and SSU; 50S and 30S in bacteria, respectively). In bacteria, translocation is mediated by a highly conserved five-domain (DI–DV) GTPase, elongation factor G (EF-G), the mechanism of which has been examined using biochemical[2–4], structural[5–12] and single-molecule fluorescence energy transfer (smFRET) methods[13–17]. EF-G engages the leading edge of pre-translocation (PRE) ribosome complexes bearing peptidyl-tRNA cargo within the aminoacyl (A) site and deacyl-tRNA in the adjacent peptidyl (P) site to facilitate large-scale conformational changes within and between the ribosomal subunits and tRNA substrates (Fig. 1a).

Within the PRE complex, deacyl- and peptidyl-tRNAs can rapidly and spontaneously unlock from their 'classical' positions (PRE-C) after peptide-bond formation to achieve multiple 'hybrid' states (PRE-H)[18,19]. Hybrid tRNA conformations, which are achieved by independent or concerted migration of the tRNA 3′-CCA termini to adjacent LSU-binding sites[15] coupled to a global SSU rotation[18,20], markedly lower the energetic barrier to translocation[21]. By contrast, spontaneous unlocking of the tRNA$_2$–mRNA module from the SSU is exceedingly rare[3]. Rapid translocation thus requires the action of EF-G, but how EF-G engages the dynamic PRE complex is actively debated.

Once bound to EF-G, the SSU undergoes a scissor-like conformational change in which its body and head domains rotate in opposing directions (SSU body-rotation reversal and forward head-swivel)[6,7,9,10]. SSU head-swivel carries the tRNA anticodons forward to 'chimeric hybrid' positions[9,10]. This process is intimately coupled to the sequential disengagement (unlocking) and engagement (relocking) of tRNA 3′-CCA termini and anticodon elements from LSU and SSU contacts, respectively, en route to their final post-translocation (POST) positions

in the P and exit (E) sites. The molecular basis of precise, directional tRNA$_2$–mRNA movement, and the role of EF-G-catalysed GTP hydrolysis in this process, remain incompletely understood.

To gain insight into how tRNA$_2$–mRNA movement is initiated by EF-G, and the role of GTP hydrolysis in translocation, we used smFRET to guide the capture of six cryo-electron microscopy (cryo-EM) structures of the ribosome in both early and late stages of translocation. A new early-intermediate structure stalled by the antibiotic spectinomycin (SPC) revealed that EF-G engages PRE-H ribosome complexes in an active, GTP-bound conformation to initiate unlocking of the peptidyl-tRNA cargo. The energy liberated by GTP hydrolysis thus facilitates downstream unlocking and relocking events in both subunits that ensure precise directional movement of the tRNA$_2$–mRNA module.

## smFRET-guided cryo-EM of translocation

We used smFRET to define reaction conditions that slow translocation sufficiently such that intermediate structures could be captured by cryo-EM. As previously described[13], the antibiotics SPC and fusidic acid specifically stall transitions after EF-G binding (intermediate states 1 (INT1) and 2 (INT2)), without otherwise altering the translocation reaction coordinate (Fig. 1a, b, Extended Data Fig. 1a–g). The FRET efficiency values of states sampled in the presence of SPC and fusidic acid were indistinguishable from those observed in the absence of the drugs[13]. We initiated pre-steady-state reactions using the same conditions used for smFRET before rapid (within 20 s) transfer to cryo-EM grids. This approach yielded six high-resolution (2.3–2.8 Å) ribosome structures programmed with deacyl-tRNA$^{Phe}$ and fMet-Phe-Lys-tRNA$^{Lys}$ at sequential stages of translocation (Extended Data Fig. 2, Supplementary Table 1), including the first—to our knowledge—structure of EF-G bound to a ribosome in an active conformation before inorganic phosphate

[1]Department of Structural Biology, St. Jude Children's Research Hospital, Memphis, TN, USA. [2]Tri-Institutional PhD Program in Chemical Biology, Weill Cornell Medicine, New York, NY, USA. [3]Institut für Medizinische Physik und Biophysik, Charité – Universitätsmedizin Berlin, Berlin, Germany. [✉]e-mail: Scott.Blanchard@stjude.org

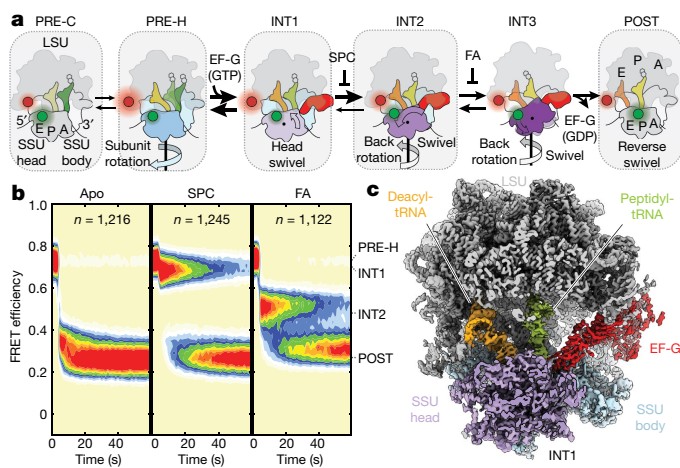

**Fig. 1 | Early kinetic and structural intermediate of tRNA₂–mRNA translocation. a**, Schematic of the translocation reaction coordinate in bacteria depicting SSU body-rotation (blue) and head-swivel (purple). tRNAs are coloured on a gradient from the A (green) to P (yellow) to E (orange) sites. The states enclosed in dashed boxes were characterized in this study. Green (donor; uS13, LD550) and red (acceptor; uL1, LD650) circles denote fluorophore positions (see **b**). FA, fusidic acid. **b**, Population FRET histograms showing FRET evolution over time upon EF-G injection with buffer, SPC (3 mM) or fusidic acid (FA, 400 μM). *n* represents the number of observed molecules. **c**, Overview of the INT1 ribosome structure captured by SPC, coloured as in **a**.

($P_i$) release (designated INT1; Fig. 1c). All structures showed density corresponding to codon–anticodon interactions, post-transcriptional tRNA modifications and a tripeptide-linked peptidyl-tRNA, indicating successful complex capture (Extended Data Fig. 3).

The POST complex, containing classical E- and P-site tRNAs (E/E, P/P) was defined as having 0° of inter-subunit rotation or SSU head-swivel (Supplementary Table 2) and the +1-mRNA position was defined as the nucleotide paired with deacyl-tRNA[Phe] position 37. The observed inter-subunit rotation, SSU head-swivel and tRNA positions (Extended Data Fig. 4, Supplementary Video 1)—together with the temporal order of conformational changes evidenced by smFRET[13] (Extended Data Fig. 1h–o, Supplementary Table 3)—were used to elucidate the molecular underpinnings of tRNA₂–mRNA translocation.

## SSU unlocking initiates spontaneously

Before EF-G engagement, the PRE-C complex (P/P, A/A) exhibited complete SSU shoulder-domain closure around the peptidyl-tRNA cargo[22,23] (Extended Data Fig. 4b). As anticipated[18,24,25], spontaneous SSU rotation during PRE-H (P/E, A/P) formation remodelled intersubunit bridges B1 and B2 and shifted the nearly universally conserved G19–C56 base pair in the deacyl-tRNA elbow domain to its fully translocated position in the E site[8,11] (Extended Data Fig. 5a–c, Supplementary Videos 2, 3). We observed two PRE-H conformations in which the peptidyl-tRNA 3′-CCA terminus paired with the LSU P site (Extended Data Fig. 5d). These states represent PRE-H2* and PRE-H1 conformations[15,16], wherein the G19–C56 pair in the peptidyl-tRNA elbow remains either fixed against the LSU A-site finger (ASF) or swings by approximately 27 Å towards the E site to engage LSU Helix 84 (H84), respectively (Supplementary Table 4). Both PRE-H conformations exhibited increased SSU body-rotation and head-swivel together with tRNA-bend angle changes (Extended Data Figs. 4, 6, Supplementary Table 2, Supplementary Video 4). Indicative of incomplete translocation on the LSU, the universally conserved, potentially catalytic LSU base A2602 was sequestered away from the peptidyl transferase centre (Extended Data Fig. 5d), which is likely to contribute to the reduced reactivities of PRE-H conformations towards the antibiotic puromycin[16,26].

Bending of the tRNA bodies enabled the tRNA anticodons and mRNA to remain in their locked SSU positions during PRE-H formation (Extended Data Fig. 7). The PRE-C–PRE-H1 transition broke SSU shoulder contact with the head domain to partially unlock the grip of the ribosome on the peptidyl-tRNA cargo (Extended Data Fig. 7a). Simultaneously, the universally conserved monitoring base G530 of the SSU shoulder disengaged from the A-site wobble pair to open the mRNA entrance channel (Extended Data Fig. 7b–d, Supplementary Video 5). This spontaneous, partial reversal of SSU domain closure was most pronounced in PRE-H1, potentially contributing to peptidyl-tRNA drop-off from PRE-H states[21,27].

## EF-G initiates peptidyl-tRNA movement

In the early translocation intermediate (INT1), the tRNA-like DIV of EF-G engaged the minor groove of the peptidyl-tRNA anticodon–mRNA codon minihelix (Fig. 2a, Extended Data Fig. 8). DIV loop II wedged between the monitoring bases of SSU helix 44 (h44; A1492 and A1493) and the codon–anticodon pair to unlock peptidyl-tRNA from the SSU A site, lifting the peptidyl-tRNA–mRNA pair out of the decoding centre (Fig. 2b, c). In contrast to later translocation stages[6,10,11], loop I in DIV interacted electrostatically with the peptidyl-tRNA phosphate backbone (Extended Data Fig. 8a, b), potentially aiding early EF-G association and positioning.

Consistent with complete SSU unlocking at the leading edge, forward peptidyl-tRNA progression tilted the SSU head away from the body (Extended Data Fig. 4b, Supplementary Video 5), extracting SSU body base C1397 from mRNA intercalation and shifting the mRNA register relative to G530 (Extended Data Fig. 7b–d). These changes flattened the kink between the A- and P-site codons[28], modestly relaxed the peptidyl-tRNA bend angle and enabled C1054 of the SSU head to pair with the +7 mRNA (Extended Data Figs. 6, 7c, Supplementary Table 4). No longer within reach of the tRNA₂–mRNA module, the A1492 monitoring base and A1913 at the tip of LSU H69 inserted into h44 to relock into their POST positions (Fig. 2c). These findings rationalize how peptidyl-tRNA fixation within the decoding centre efficiently inhibits SSU unlocking and translocation[5,29,30].

EF-G engagement had a limited effect outside of the decoding centre, maintaining the inter-subunit rotation angle and LSU positions of both tRNAs from PRE-H1. We did, however, observe the formation of interactions between the SSU body and the deacyl-tRNA anticodon–mRNA codon pair (Extended Data Fig. 9), which is consistent with an allosteric securing of the reading frame in the E site.

## EF-G engages in an active conformation

Coincident with DIV-mediated peptidyl-tRNA unlocking from the decoding centre of the SSU, the G domain (DI) of EF-G packed intimately against the catalytic sarcin–ricin loop (SRL) to rigidify the GTPase-activating centre and shift it away from the SSU (Supplementary Video 2). To bridge the gap between the decoding centre and the GTPase-activating centre, EF-G adopted an elongated conformation (Fig. 3a, Extended Data Fig. 8c).

Unexpectedly, the G domain contained strong, continuous electron density for α, β and γ phosphates at all thresholds, supporting the presence of a GTP molecule in the nucleotide-binding pocket (Fig. 3b, Extended Data Fig. 10). Congruent with a pre-hydrolysis state, smFRET experiments revealed that INT1 transit was markedly slowed by the non-hydrolysable GTP analogue GTPγS (Extended Data Fig. 1p–t). Although we cannot unambiguously determine whether EF-G is bound to GTP, GDP-$P_i$ or a mixture of the two in dynamic exchange[31], we can conclude that EF-G is capable of unlocking the peptidyl-tRNA cargo from the decoding centre of the SSU before $P_i$ release. Hence, although pre-hydrolysis EF-G conformations have been trapped on substrates that lack peptidyl-tRNA cargo or on POST complexes using

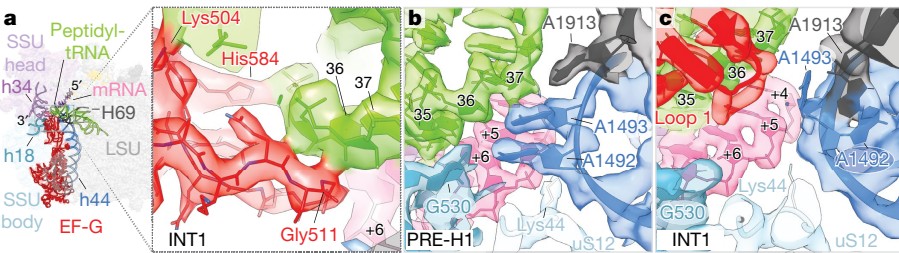

**Fig. 2 | Unlocking of the peptidyl-tRNA decoding centre. a**, Locally filtered electron density illustrating shape-specific recognition of the A-site codon–anticodon pair by EF-G (red) in its active, GTP-bound conformation (INT1). **b**, **c**, Unlocking of the tRNA$_2$–mRNA decoding centre in the PRE-H1 (**b**) to INT1 (**c**) transition. Peptidyl-tRNA, green; mRNA, pink; H69, grey; h44, blue; h18, cyan; uS12, light blue. Threshold $\sigma = 5$.

non-hydrolysable GTP analogues[8,9,32], or using a catalytically dead EF-G mutant (H92A)[12], the INT1 structure captured here represents the best approximation to date of EF-G bound to its physiological substrate in its active, GTP-bound conformation.

Consistent with an active GTP conformation[33], the switch-I and catalytic switch-II elements were fully structured to encircle the guanosine nucleotide (Fig. 3c). As observed for G domains of other GTP-bound TRAFAC-family GTPases[33], the switch-I, switch-II and P-loop regions engaged the β and γ phosphates via $Mg^{2+}$ coordination. The catalytic switch-II residue His92 was also positioned 4 Å from the γ phosphate, primed to facilitate GTP hydrolysis (Extended Data Fig. 10a).

In agreement with mutation sites conferring SPC resistance[34], we observed density for all three SPC rings immediately beneath the SSU P site, approximately 100 Å from the GTP-binding site[35,36] (Extended Data Fig. 11, Supplementary Video 6). Within its physiological INT1 substrate, the methyl substituent of SPC ring C stabilized the interaction of Lys26 of uS5 with h28—an interaction that is likely to prevent further SSU head-swivel at this specific stage of translocation[13,35,36].

### EF-G engages the rotated ribosome

In its active conformation, the switch-I element of EF-G exhibited a continuous, extended architecture that bridged the G domain with DII and DIII (Fig. 3c). This region is disordered in nearly all EF-G structures both on and off the ribosome, with the exception of an isolated crystal structure of a thermophilic EF-G homologue (EF-G-2) bound to GTP[37] (Extended Data Fig. 8c) and structures of EF-G(H92A) bound to POST ribosomes[12]. The switch-I N terminus interacted with both the rotated SSU body and the LSU, anchoring His38 on the intersubunit bridge B8

fidelity determinant[38] and extending by approximately 19 Å to contact the SRL (Fig. 3c). Because switch-I ordering is contingent on the precise distance between these ribosomal elements, we posit that these stabilizing contacts provide the energy needed for EF-G–GTP binding to unlock the peptidyl-tRNA cargo from the SSU to initiate translocation.

Similar to the structures of GTP-bound EF-G-2[37] and EF-G(H92A)[12], the extended switch-I structure nucleated a modified β-barrel fold in DII (Fig. 3c, Extended Data Fig. 8d), suggesting that DII has an intramolecular effector role[39]. This non-canonical DII architecture mediated EF-G contact with the conserved U368–A55 tertiary pair where the SSU shoulder and body domains diverge, a region that has been implicated in activating GTP hydrolysis on elongation factor Tu (EF-Tu) during tRNA selection[40]. The modified β-barrel fold also buttressed the switch-I C terminus against the highly conserved DIII helix B3[9,12,32] (Fig. 3c), providing a conduit for information transfer from the SSU shoulder–body interface to the G domain of EF-G. Because this network of contacts is specifically underpinned by interactions with the rotated SSU, we propose that the activation of GTP hydrolysis in EF-G is triggered by formation of the extended switch-I fold or by changes in the SSU rotation angle during later steps of translocation.

### P$_i$ release remodels the conformation of EF-G

By comparing INT1 with the structure of INT2 stalled by fusidic acid, we obtained additional insights into the role and timing of GTP hydrolysis by EF-G. As anticipated[6,10], we observed loss of density for the nucleotide γ phosphate and switch I, and a restoration of the canonical DII β-barrel fold in the INT2 complex (Extended Data Figs. 8d, 10d). These post-hydrolysis changes correlated with an upward shift and an approximately 15° rotation of the G domain of EF-G relative to the SRL (Extended Data Fig. 8e), together with inward displacement of the entire GTPase-activating centre towards the LSU central protuberance (Supplementary Video 2). Despite such extensive remodelling, EF-G DIV loops I and III remained in direct contact with the peptidyl-tRNA anticodon–mRNA codon pair, while losing contact with the tRNA body[6,9,10] (Extended Data Fig. 8b). Consequently, all five EF-G domains reached further into the inter-subunit space, coupled with an approximately 17° hinge-like motion between DIV and DV roughly perpendicular to the SSU interface (Extended Data Fig. 8f, Supplementary Video 7).

As expected[6,10], the altered position and conformation of EF-G in INT2 was associated with a scissor-like reverse rotation of the SSU body towards its POST position and forward SSU head-swivel in the direction of translocation (Extended Data Fig. 4b). Such changes collapsed the SPC-binding pocket (Extended Data Fig. 11c), while establishing direct contact between DIV and the SSU head domain[6,9,10] (Extended Data Fig. 8b, d) and a new intersubunit bridge involving the ASF, uS19 and the LSU central protuberance[7,10] (Extended Data Fig. 5c), potentially stabilizing the head-swivel angle. The observed scissor-like conformational changes were reduced in amplitude compared with those found in previous investigations[6,10], which probably reflects the diffusive nature of SSU head- and body-domain motions and their sensitivity

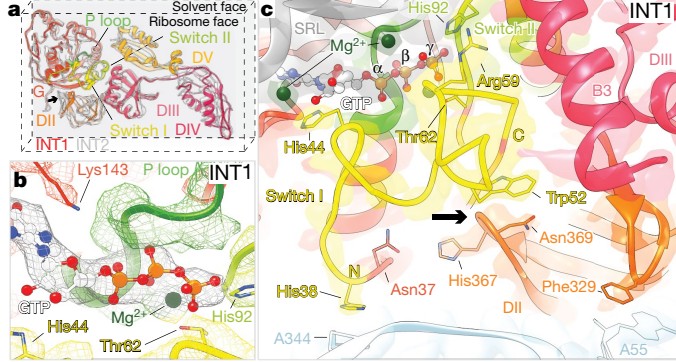

**Fig. 3 | Overview of the active, GTP-bound conformation of EF-G. a**, Domain architecture of EF-G in its active, GTP-bound conformation (INT1, coloured) and in a post-hydrolysis conformation (INT2, grey, G-domain alignment). **b**, Locally filtered electron density (mesh) in the nucleotide-binding pocket for INT1. **c**, Elongated switch-I (residues 38–68, yellow) contacts with the SRL (grey), the SSU (light blue), DII (orange) and DIII (pink). The conformational change of DII is indicated with an arrow. Threshold $\sigma = 6$.

to ribosome composition and/or experimental condition[13,41]. We infer from these observations that entrance into the INT2 basin liberates a range of intersubunit rotation and SSU head-swivel angles[13]—and related conformational processes in EF-G and the ribosome—that can facilitate GTP hydrolysis and/or $P_i$ release.

## Head-swivel initiates deacyl-tRNA movement

The INT1–INT2 transition moved the entire tRNA$_2$–mRNA module by approximately 8.5 Å towards its POST position (Extended Data Fig. 4c, Supplementary Table 2), enabled in part by the maintenance of anchored stacking interactions between deacyl- and peptidyl-tRNA and the SSU head (Extended Data Fig. 12). Movement of the deacyl-tRNA anticodon triggered release of the C1400 base from the deacyl-tRNA anticodon–mRNA codon pair and disrupted E-site mRNA codon stacking with the SSU 690 loop (Extended Data Fig. 12d, e). Notably, only two of the three E-site codon nucleotides shifted relative to the G926 fiducial marker and the mRNA exit channel (Extended Data Fig. 12), establishing that the tRNA$_2$–mRNA module is only partially translocated with respect to the SSU body.

The INT1–INT2 transition also unlocked the interface between uS7 and uS11 (Extended Data Fig. 9) and widened the gap between the L1 stalk and the SSU head at the lagging edge. Simultaneously, the peptidyl-tRNA G19–C56 elbow pair and the A2602 base of the LSU relocked into their fully translocated positions (Extended Data Fig. 5c, d). Movement of the tRNA$_2$–mRNA module also relocked SSU bases C1397 and A1493 on the leading edge into their POST positions, intercalated on opposite sides of the downstream mRNA codon[42] (Extended Data Fig. 7b–d). The INT1–INT2 transition therefore completes relocking events on the LSU and in the SSU decoding centre while mediating a distinct SSU unlocking process at the lagging edge of the ribosome. Such changes are likely to contribute to reading frame maintenance while opening pathways through which deacyl-tRNA can shift position and/or dissociate[13,17,42].

## Discussion

Although snapshots of translocation have been previously reported[6–11], structural information on the initial engagement of GTP-bound EF-G with its physiological substrate has been missing. Our structures reveal persistent engagement of the peptidyl-tRNA cargo during the relay of tRNA$_2$–mRNA module unlocking and relocking events on both ribosomal subunits. The sequential unlocking mechanism observed is initiated by PRE-complex dynamics. EF-G engages spontaneously achieved PRE-H conformations in its active, GTP-bound conformation, unlocking the decoding centre and sending the peptidyl-tRNA on an arc-like trajectory in single-nucleotide increments (Fig. 4), as initially inferred from optical trapping studies of mRNA unwinding[43]. By contrast, deacyl-tRNA movement is not initiated during unlocking at the decoding centre but is instead coordinated by a second SSU unlocking process at the lagging edge, which enables a coupled shift of the entire tRNA$_2$–mRNA module in the INT1–INT2 transition (Fig. 4). Notably, translocation also involves the formation of POST-like contacts in both ribosomal subunits (relocking events), which may provide a thermodynamic driving force for forward progression while securing the translation reading frame.

Non-competitive elongation-factor binding to the ribosome stipulates distinct recognition features. Our findings support a parsimonious model in which EF-G preferentially engages rotated PRE-H conformations[16], whereas EF-Tu recognizes the locked, unrotated ribosome. This model avoids steric clashes between EF-G and constituents of the PRE-C complex and ensures that peptide-bond formation and LSU unlocking have occurred before energy expenditure. Rotated ribosome conformations are also expected to stabilize the extended switch-I structure and the modified DII fold of EF-G, which are likely to be prerequisites

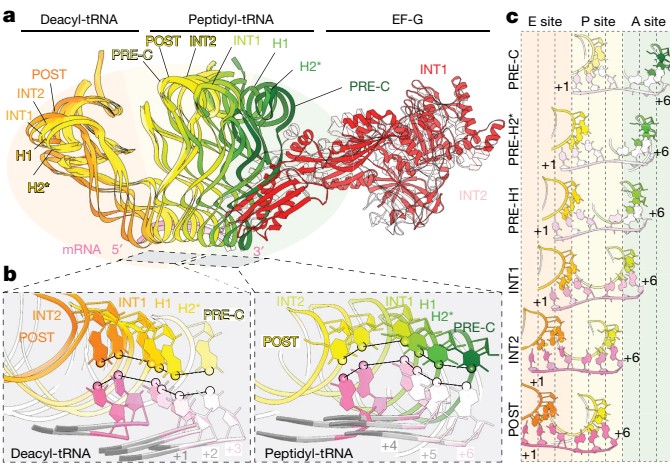

**Fig. 4 | Non-uniform tRNA$_2$–mRNA movement during translocation.**
**a**, Overlay of the tRNA$_2$–mRNA module from the A (green) to P (yellow) to E (orange) sites. **b**, Overlay from **a**, viewed from the codon–anticodon interface. Circles on the tRNAs at position 34 N1 (deacyl-tRNA, left) and N3 (peptidyl-tRNA, right) depict the tRNA trajectories during translocation. **c**, tRNA anticodon–mRNA codon movement during translocation, same perspective as **b**.

for activation of GTP hydrolysis. Such a model helps to explain how PRE-H states lower the energetic barrier to translocation[21]; the unexpectedly high Michaelis constant of EF-G-catalysed translocation; and the dependency of translocation rate on PRE complex composition[3,4,13]. Structural and mechanistic conservation posit that this division of elongation-factor recognition may extend across domains of life.

Although our findings provide information on the molecular basis of early-translocation events, analogous strategies will need to be applied to late-translocation processes, during which deacyl- and peptidyl-tRNA unlock from the SSU head to progress to the POST state (Extended Data Fig. 12). Such events putatively include exaggerated swivel-like motions of the SSU head and relocking of peptidyl-tRNA in its ultimate P-site position[13,17]. Combined structural, smFRET and molecular-dynamics studies will also be vital in defining the precise timing of GTP hydrolysis, $P_i$ release and the dissociation of GDP-bound EF-G. Delineation of the complete translocation mechanism will provide a deeper understanding of the regulation of translation, including the programmed errors that govern normal physiology and disease[44].

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

## Methods

### Data reporting

No statistical methods were used to predetermine sample size. The experiments were not randomized and the investigators were not blinded to allocation during experiments and outcome assessment.

### Buffers and reagents

All experiments were carried out in either polymix buffer A (50 mM Tris-OAc (pH 7.5), 100 mM KCl, 5 mM NH$_4$OAc, 0.5 mM Ca(OAc)$_2$, 5 mM Mg(OAc)$_2$, 6 mM 2-mercaptoethanol, 0.1 mM EDTA, 5 mM putrescine and 1 mM spermidine)[45] or polymix buffer B (30 mM HEPES pH 7.5, 5 mM MgCl$_2$, 50 mM NH$_4$Cl, 5 mM 2-mercaptoethanol, 2 mM spermidine and 5 mM putrescine)[46]. A cocktail of triplet-state quenchers (1 mM Trolox, 1 mM nitrobenzyl alcohol and 1 mM cyclooctatetraene) and an enzymatic oxygen scavenging system (protocatechuic acid (PCA)/ protocatechuate-3,4-dioxygenase (PCD)) were used for smFRET experiments. Spectinomycin sulfate was purchased from MP Biomedicals. Fusidic acid sodium salt, GTP and GTPγS were from Sigma-Aldrich. GTP was further purified using a Mono Q 5/50 GL anion exchange column (GE Healthcare Life Sciences). Pyruvate kinase, myokinase and phosphoenolpyruvate (PEP) were purchased from Sigma-Aldrich. All other standard reagents were purchased from Sigma-Aldrich or VWR.

### Cryo-EM and smFRET sample preparation

**Purification of ribosomes and elongation factors.** Wild-type, uS13- and uL1-labelled ribosomal subunits were purified from *Escherichia coli* BL21 and MRE600 for smFRET and cryo-EM experiments, respectively, as previously described[13,45,47]. EF-Tu[48] and EF-G[15] were purified as previously described. *E. coli* tRNA$^{fMet}$, tRNA$^{Phe}$ and tRNA$^{Lys}$ were purified[13,16,24] and tRNA$^{Phe}$ was labelled with LD655 at the acp$^3$ modification on nucleotide U47, as described previously[45]. Wild-type, uS13- and uL1-labelled initiation complexes were prepared as previously described[45,47,49].

**Preparation of ternary complex for smFRET experiments.** Phenylalanine (2.5 mM), PheRS (0.15 µM), pyruvate kinase (0.4 µM), myokinase (0.5 µM), PEP (3.75 mM), GTP (630 µM) and LD655-labelled tRNA$^{Phe}$ (250 nM) were combined in charging buffer (50 mM Tris pH 8, 10 mM KCl, 100 mM NH$_4$Cl, 10 mM MgCl$_2$, 1 mM DTT, 2.5 mM ATP and 0.5 mM EDTA) before addition of EF-Tu–EF-Ts (EF-Ts, elongation factor thermostable) (1 µM). The resulting mixture was incubated for 10 min at 37 °C to aminoacylate the tRNA (aa-tRNA) and form a ternary complex (EF-Tu–aa-tRNA–GTP). Before injection into the microscope flow cell for smFRET imaging, ternary complex was diluted 40× (to a final concentration of 6 nM) in imaging polymix buffer containing 0.5 mM GTP.

**Preparation of Phe-tRNA$^{Phe}$ ternary complex for cryo-EM experiments.** Phenylalanine (1 mM), PheRS (0.2 µM), pyruvate kinase (0.6 µM), myokinase (0.6 µM), PEP (0.4 mM), GTP (1 mM) and tRNA$^{Phe}$ (1.6 µM) were combined in charging buffer before addition of EF-Tu–EF-Ts (8 µM). The resulting mixture was incubated for 10 min at 37 °C to aminoacylate the tRNA and form a ternary complex. Successful aminoacylation was confirmed by fast protein liquid chromatography (FPLC).

**Preparation of Lys-tRNA$^{Lys}$ ternary complex for cryo-EM experiments.** Lysine (1 mM), LysRS (0.6 µM), pyruvate kinase (0.6 µM), myokinase (0.6 µM), PEP (0.4 mM), GTP (1 mM) and tRNA$^{Lys}$ (3 µM) were combined in charging buffer and incubated for 15 min at 37 °C to aminoacylate the tRNA. EF-Tu–EF-Ts (15 µM) was added to the mixture and incubated for 5 min at 37 °C to form a ternary complex. Successful aminoacylation was confirmed by FPLC.

**Preparation of elongator POST complexes for cryo-EM.** All reactions were performed in the presence of a GTP regeneration system[50]. Initiation complexes at a concentration of approximately 3 µM were prepared with MFK mRNA (Biotin-5′-CAA CCU AAA ACU UAC ACA CCC UUA GAG GGA CAA UCG **AUG UUC AAA** GUC UUC AAA GUC AUC-3′) and fMet-tRNA$^{fMet}$ in the P site. mRNA nucleotide position 40 corresponds to the +1 position. Initiation complexes were incubated with ternary complex containing Phe-tRNA$^{Phe}$ (around 1.6 µM) for 5 min at 37 °C to form the PRE translocation complex. PRE complexes were incubated with sub-stoichiometric concentrations of GTP-bound EF-G (300 nM) for 10 min at 37 °C to form the elongator POST translocation complex (fMet-Phe-tRNA$^{Phe}$ in the P site). Additional reagents were added to the mixture to aminoacylate free tRNA in solution. Elongator POST complexes were pelleted over a 37% sucrose cushion containing buffer A at 437,000*g* in a TLA-100.3 rotor (Beckman) for 4 h at 4 °C to remove EF-G and deacyl-tRNA. Pelleted elongator complexes were resuspended in buffer A for a final concentration of 9 µM and were flash-frozen.

**Preparation of elongator complexes for cryo-EM.** Elongator POST complexes containing fMet-Phe-tRNA$^{Phe}$ in the P site were thawed and diluted in buffer B with 1 mM GTP for a final concentration of 2 µM ribosomes. For preparation of the translocation intermediate samples, SPC (INT1) or fusidic acid (INT2) were added to the dilution buffer. SPC was used at its half-maximum inhibitory concentration (IC$_{50}$) for translocation inhibition (3 mM)[51]. Fusidic acid was used at near-saturating concentration (400 µM)[13,52]. The elongator POST complexes were incubated with Lys-tRNA$^{Lys}$ ternary complex (2 µM final) for around 30 s at 25 °C to fill the A site. The resulting elongator PRE complex (P-site tRNA$^{Phe}$; A-site fMet-Phe-Lys-tRNA$^{Lys}$) was either added to cryo-EM grids directly (PRE) or incubated with EF-G (5 µM final) in the absence (POST) or presence of SPC (INT1) or fusidic acid (INT2) for around 5–10 s before the solution was applied to cryo-EM grids.

**Cryo-EM grid preparation.** Cryo-EM grids were prepared using a Vitrobot Mark IV plunge-freezing device (Thermo Fisher Scientific). For each experiment, 3 µl of sample was applied to Quantifoil R1.2/1.3 holey carbon Cu 300 mesh (INT1 and INT2) or Au 300 mesh (PRE and POST) grids that had been glow-discharged (Ar/O$_2$) for 20 s using a Solarus II Plasma Cleaning system (Gatan). Grids were incubated in the Vitrobot chamber for 10 s at 10 °C at 95% humidity before blotting (6 s; blot force −5) and plunge freezing into liquid ethane.

### smFRET imaging of translocation

Ribosomes programmed with 5′-biotinylated mRNA substrates containing P-site-bound fMet-tRNA$^{fMet}$ and displaying the codon UUC in the A site were immobilized on passivated coverslips as described previously[13,45]. The ribosomes were then incubated for 2 min with ternary complex containing either LD655-labelled Phe-tRNA$^{Phe}$ or unlabelled Phe-tRNA$^{Phe}$, leading to stoichiometric formation of either PRE ribosomes containing A-site LD655-labelled fMet-Phe-tRNA$^{Phe}$, P-site tRNA$^{fMet}$ and LD550-labelled uS13 or PRE ribosomes containing A-site fMet-Phe-tRNA$^{Phe}$, P-site tRNA$^{fMet}$ and LD550-labelled uS13 and LD650-labelled uL1. To initiate translocation, EF-G with either 1 mM GTP or 1 mM GTPγS, with or without 3 mM SPC or 400 µM fusidic acid, was delivered to the flow cell by stopped-flow injection. All smFRET experiments were carried out at 25 °C. The time-evolution of the FRET signal was then recorded using a home-built total-internal-reflection-based fluorescence microscope[53] with laser (532 nm) illumination at 0.1 kW cm$^{-2}$ at a time resolution of 40 or 400 ms. Donor and acceptor fluorescence intensities were extracted from the recorded movies and FRET efficiency traces were calculated using custom software implemented in MATLAB R2015b. FRET traces were selected for further analysis according to the following criteria: a single catastrophic photobleaching event; at least 8:1 signal-to-background-noise ratio and 6:1 signal-to-signal/noise ratio; less than four donor–fluorophore blinking events; a correlation coefficient between donor and acceptor <0.5. The resulting smFRET traces were analysed using hidden Markov model idealization methods as implemented in the SPARTAN software

package (v.3.7.0)[53]. In all idealizations, transitions between all states were allowed. The model used for uS13 to peptidyl-tRNA FRET had four states (FRET values: $0.14 \pm 0.04$; $0.30 \pm 0.03$; $0.50 \pm 0.05$; $0.75 \pm 0.06$); the model for uS13 to uL1 FRET had three FRET states (FRET values: $0.73 \pm 0.05$; $0.48 \pm 0.08$; $0.27 \pm 0.04$). To compare the translocation kinetics under different conditions from the idealized FRET traces, we constructed normalized cumulative distributions over the arrival time to the POST state, defined as the 0.50 FRET state for the uS13 to peptidyl-tRNA signal and the 0.27 FRET state for the uS13 to uL1 signal.

## Cryo-EM data collection
Cryo-EM data were collected using a Titan Krios G3i (Thermo Fisher Scientific) transmission electron microscope equipped with a K3 direct electron detector and post column GIF (energy filter). K3 gain references were acquired just before data collection. Data collection was performed using SerialEM software (v.3.7.1)[54] with image shift protocol (nine images were collected with one defocus measurement per nine holes). Movies were recorded at defocus values from −0.5 μm to −1.5 μm at a magnification of 105,000×, which corresponds to the pixel size of 0.826 Å per pixel at the specimen level (super-resolution 0.413 Å per pixel) for the apo PRE, POST and INT1 structures. During the 2.4-s exposure, 60 frames (0.04 s per frame, 1.4596 $e^-$ per frame per Å$^2$) were collected with a total dose of around 87 $e^-$ per Å$^2$. The first frame was discarded. Motion correction was performed on raw super-resolution movie stacks and binned twofold using Motion-Cor2 software[55]. Cryo-EM data for the INT2 complex was collected at a magnification of 82 kx (1.06 Å per pixel; super-resolution 0.53 Å per pixel), with a total dose of around 70 $e^-$ per Å$^2$. CTF parameters were determined using CTFFind4[56] and refined later in Relion[57] (v.3.1) and cryoSPARC[58] (v.3). Before particle picking, good micrographs were qualified by power spectrum. Particles were picked using cisTEM[59] and the coordinates were transferred to Relion (see below for details of classification and refinement). Sharpened and locally filtered maps were used to aid in model building. Electron density map values were normalized to mean = 0 and standard deviation ($\sigma$) = 1 in UCSF Chimera using the vop scale function. For detailed information on data collection parameters and model-building statistics see Extended Data Fig. 2 and Supplementary Table 1.

## Cryo-EM data processing for the apo PRE structures
Prior to particle picking, good micrographs were qualified by power spectrum (7,183 movie stacks). Particles were picked within cisTEM (659,777 particles). After extraction in Relion (fourfold binned), several rounds of the 2D classification were performed in cryoSPARC. An Ab initio structure was built in cryoSPARC and then used as a reference for 3D classification in Relion. Particles from good classes (534,348 particles) were then re-extracted (twofold binned) and refined in Relion followed by CtfRefine and 3D classification into 10 classes. Class 4 possessed an unrotated SSU (225,423 particles) and class 9 possessed a rotated SSU (160,291 particles). Unrotated SSU and rotated SSU classes were individually subjected to 3D refinement in Relion and sorted further into 5 classes using 3D classification. From 3D classification of the unrotated particles, two classes contained classical A- and P-site tRNAs, which were combined (109,769 particles) and run through 3D refinement (un-binned) yielding the PRE-C structure. From 3D classification of the rotated particles, one class showed evidence of hybrid P-site tRNA and classical A-site tRNA (PRE-H2*; 33,330 particles). Two classes from the unrotated 3D classification contained weak density for A-site tRNA, which were combined (126,699 particles) and further classified with an A-site mask to improve ligand density. From this focused classification of the A-site tRNA, one class contained hybrid deacyl-tRNA and peptidyl-tRNA (PRE-H1; 51,685 particles). After 3D classification, particles from PRE-C, PRE-H2* and PRE-H1 classes were re-extracted with the full pixel size and refined in Relion according to the gold-standard criteria.

## Cryo-EM data processing for the apo POST structure
Before particle picking, good micrographs were qualified by power spectrum (2,834 movie stacks). Particles were picked within cisTEM (439,422 particles). After extraction in Relion (fourfold-binned), particles were refined in Relion followed by CtfRefine and 3D classification into 10 classes. Five classes possessed an unrotated SSU, which were combined (120,513 particles), re-extracted and refined in Relion (twofold-binned). To improve the density for E-site tRNA, we performed a focused 3D classification using an E-site mask. This yielded two classes with solid E-site tRNA density, which were combined (34,688 total particles), re-extracted with the full pixel size and refined in Relion according to the gold-standard criteria for the apo POST complex structure.

## Cryo-EM data collection and processing for the INT1 structure
Before particle picking, good micrographs were qualified by power spectrum (11,916 movie stacks). Particles were picked within cisTEM (1,001,439 particles). After extraction in Relion (fourfold-binned), several rounds of the 2D classification were performed in cryoSPARC. Particles from good 2D classes (652,128 particles) were then refined in Relion and sorted using 3D classification into 6 classes. One class possessed a rotated SSU (184,857 particles), which was refined and further classified in Relion into 8 classes. One of these classes contained EF-G (33,688 particles). These particles were re-extracted with the full pixel size and refined in Relion according to the gold-standard criteria for the INT1 complex structure.

## Cryo-EM data collection and processing for the INT2 structure
Before particle picking, good micrographs were qualified by power spectrum (6,651 movie stacks). Particles were picked within cisTEM (1,259,307 particles). After extraction in Relion (fourfold-binned), several rounds of 2D classification were performed in cryoSPARC. Particles from good 2D classes (639,984 particles) were then refined in Relion and sorted using 3D classification into 6 classes. One class possessed a rotated SSU with EF-G bound (113,540 particles), which was refined in Relion. To improve occupancy of the tRNAs and EF-G, we performed a focused 3D classification using a ligand mask into three classes in Relion. One class contained two tRNAs and EF-G (33,008 particles). These particles were re-extracted with the full pixel size and refined in Relion according to the gold-standard criteria for the INT2 complex structure.

## Molecular model building
Models of 50S (starting model PDB ID: 4YBB[60]), 30S (starting model PDB ID: 4YBB[60]), tRNA$^{Lys}$ (starting model PDB ID: 5E81[61]), tRNA$^{Phe}$ (starting model PDB ID: 4WRO[62]), EF-G (starting model PDB ID: 4V9O[32]) and ribosomal protein L7/L12 (starting model PDB ID: 1CTF[63]) were fitted into EM maps and refined through iterative rounds of manual model building in Coot (v.0.9.4.1)[64], refinement of RNA with ERRASER[65] and real-space refinement using Phenix (v.1.19-4092)[66]. mRNA nucleotide 40 corresponds to the +1 position. The nascent peptide and mRNA were built de novo in Coot. ATP molecules were modelled between 23S (1) U369 and A404 and (2) U40 and U441. An ATP was also modelled in the LSU E site for PRE-C. Polyamines were modelled into tubular unassigned density displaying the appropriate surrounding electrochemical environment. Notably, putrescine molecules were modelled in the E site of PRE-H2* and PRE-H1 proximal to 16S A790. The acp$^3$ modification on U47 of tRNA$^{Phe}$ and tRNA$^{Lys}$ was also modelled de novo as follows: the 3-amino-3-carboxypropyl moiety was added to position 3 of the pyrimidine ring of uridine monophosphate, saved as a novel modified RNA-nucleotide acp$^3$U with ligand code 3au. Restraints for refinement were generated using phenix.elbow[67]. Models were validated using phenix.validation_cryoem[68] with the built-in MolProbity[69] scoring. See Supplementary Table 1 for more information. In each complex, we also observed fragmented electron density for the Shine–Dalgarno-like/

Anti-Shine–Dalgarno minihelix, which was modelled in PRE-C, as well as ribosomal protein uS1.

## Figure preparation

Molecular graphics and analyses were performed with UCSF Chimera[70] or ChimeraX, developed by the Resource for Biocomputing, Visualization, and Informatics at the University of California, San Francisco, with support from NIH P41-GM103311. Unsharpened maps from Relion Refine3D were used for figure images, threshold $\sigma = 6$, unless otherwise stated. Angle and distance measurements were performed in UCSF Chimera using the Fit in Map and the Distance tools. All figures were prepared using structures and models aligned on the LSU core, unless otherwise noted. The LSU core used was simulated 3 Å density of high-resolution ribosome crystal structure PDB ID: 4YBB[60] in UCSF Chimera (molmap) with the following mobile elements omitted: uL5, uL6, uL9, uL10, uL11, uL120, uL31, H34 (709–723), the A-site finger (ASF; H38; 866–906), the L11 stalk (1045–1112), H69 (1908–1925), the L1 stalk (2093–2198), H83/84 (2297–2318), the SRL (2651–2667) and 5S. Root mean square deviation (r.m.s.d.) heat maps were prepared in UCSF Chimera using the Matchmaker tool for proteins and nucleic acids. Rotation angles and axes illustrations were generated using the Measure Rotation tool in UCSF Chimera. Electron density was coloured using the Colour Zone tool with a 3 Å radius. Figures were compiled in Adobe Illustrator (Adobe). The mRNA kink in Supplementary Table 2 was measured between mRNA positions +3 (C1' of the last nucleotide of the deacyl-tRNA codon) and +4 (C1' of the first nucleotide of the peptidyl-tRNA codon).

## Reporting summary

Further information on research design is available in the Nature Research Reporting Summary linked to this paper.

## Data availability

PDBs and cryo-EM 3D maps for all structures are available through the Protein Data Bank (PDB) and the Electron Microscopy Data Bank (EMDB), respectively, as follows: PRE-C, 7N1P, EMD-24120; PRE-H2*, 7N30, EMD-24135; PRE-H1, 7N2U, EMD-24133; INT1-SPC, 7N2V, EMD-24134; INT2-FA, 7N2C, EMD-24132; POST, 7N31, EMD-24136.

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

**Acknowledgements** We acknowledge support from the Cryo-Electron Microscopy and Tomography Center, the High-Performance Computing Center and the Single-Molecule Center at St. Jude Children's Research Hospital. We also acknowledge guidance from D. Miller and J. Bollinger from the St. Jude Biomolecular X-ray Crystallography Center for their input and support of our molecular-model building and computational infrastructure. We thank D. Terry, R. Kiselev and other members of the Blanchard laboratory for their expertise and efforts to enable the single-molecule investigations performed and for their review of the manuscript during the preparation and completion of this research. This work was supported by grant funding from the National Institutes of Health (GM079238) to S.C.B., the NIH T32 (GM115327-Tan) to E.J.R., the Swedish Research Council (2017-06313) to M.H. and the German Research Foundation (DFG FOR 1805) to C.M.T.S.

**Author contributions** E.J.R. and R.B.A. prepared the biochemical components used for this investigation. A.G.M. and E.J.R. prepared cryo-EM grids, and collected and processed cryo-EM data. E.J.R., M.S and S.K.N. built and refined molecular models. M.H. performed smFRET experiments. E.J.R., S.C.B., M.S. and C.M.T.S. analysed structural data. All authors wrote the manuscript.

**Competing interests** S.C.B. and R.B.A. hold equity interests in Lumidyne Technologies. The remaining authors declare no competing interests.

**Additional information**
**Correspondence and requests for materials** should be addressed to S.C.B.

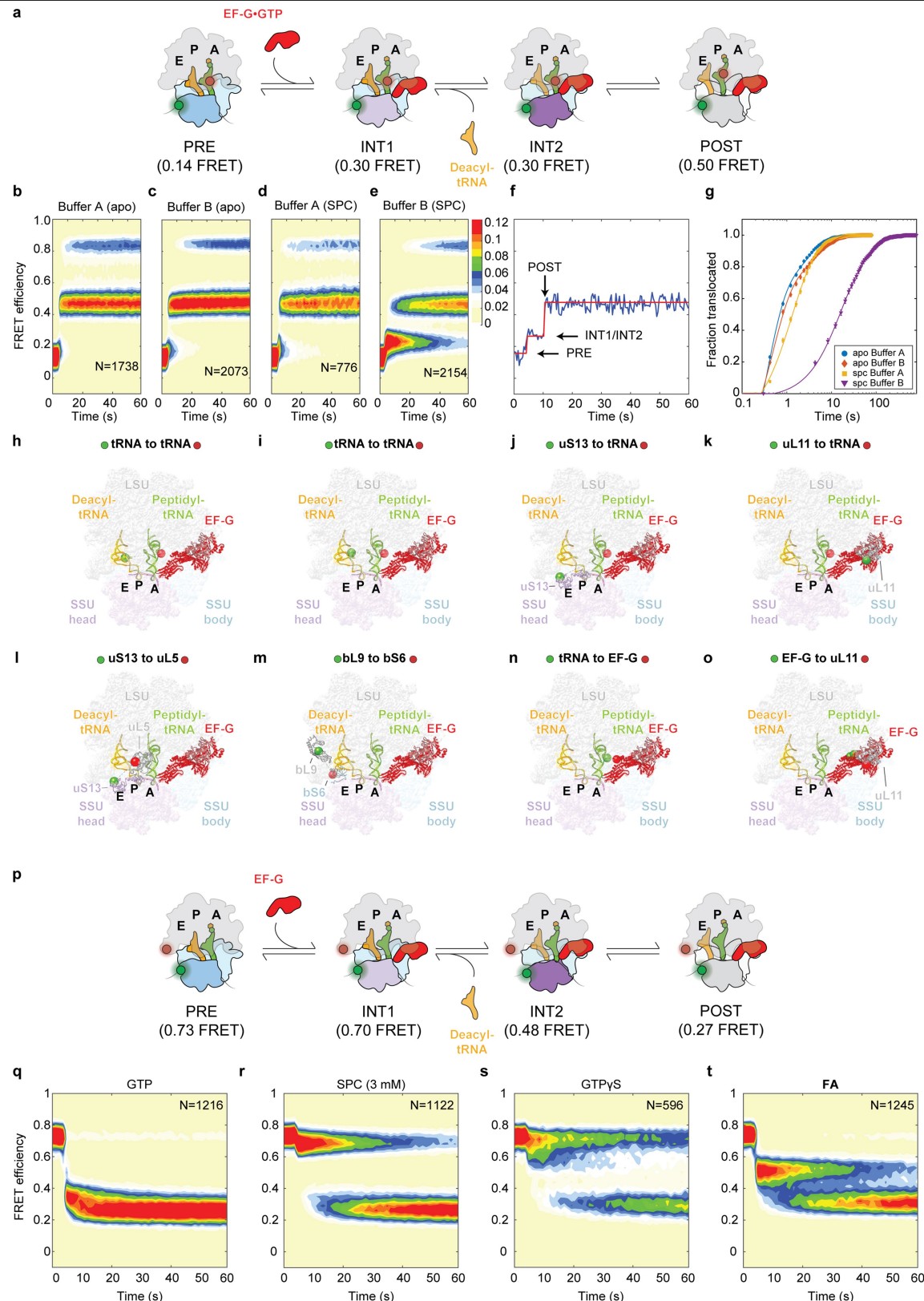

**Extended Data Fig. 1** | See next page for caption.

**Extended Data Fig. 1 | smFRET investigations of translocation. a–g,** smFRET data on translocation inhibition by SPC. **a,** Schematic of translocation indicating the positions of the donor (LD550, uS13 N terminus) and acceptor (LD655, peptidyl-tRNA U47) fluorophores, and FRET efficiency values for the indicated states. **b–e,** Population FRET histograms showing time evolution of FRET between uS13 and peptidyl-tRNA at a time resolution of 400 ms on delivery of 5 μM EF-G in either buffer A (**b, d**) or B (**c, e**), without (**b, c**) or with 3 mM SPC (**d, e**). *N* indicates the number of observed molecules. **f,** Example smFRET trace from the data in **b–e**. EF-G, injected approximately 4 s after the start of data acquisition, rapidly binds to the PRE complex (0.14 FRET), converting it into INT1, and subsequently INT2 (both 0.3 FRET), before achieving the POST state (0.5 FRET). **g,** Kinetic analysis of data as in **b–e**, demonstrating an approximately tenfold potentiating effect of buffer B on SPC inhibition. The points represent mean accumulation of translocated ribosomes with time, defined as reaching the 0.5 FRET state. Solid lines represent fits of bi-exponential functions to the data. Error bars indicate standard deviations derived from three technical experimental replicates. **h–o,** Approximate positions of published donor (green sphere) and acceptor (red sphere) dyes on the INT1 structure. **h,** Donor tRNA[Phe] (Cy3, s[4]U8), acceptor fMet-Phe-Lys-tRNA[Lys] (Cy5, acp[3]U47)[13,15]. **i,** Donor tRNA[Phe] (Cy3, acp[3]U47), acceptor fMet-Phe-Lys-tRNA[Lys] (Cy5, acp[3]U47)[13]. **j,** Donor uS13 (LD550, N-terminal ACP), acceptor fMet-Phe-Lys-tRNA[Lys] (Cy5, acp[3]U47)[13]. **k,** Donor uL11 (Cy3, residue 87), acceptor fMet-Phe-Lys-tRNA[Lys] (Cy5, acp[3]U47)[71]. **l,** Donor uS13 (LD550, N-terminal ACP), acceptor uL5 (LD650, N-terminal ACP)[13]. **m,** Donor bL9 (Cy3, residue N11C), acceptor bS6 (Cy5, residue D41C)[20]. **n,** Donor fMet-Phe-Lys-tRNA[Lys] (Cy3, acp[3]U47), acceptor EF-G (Cy5, C-terminal SFP)[72]. **o,** Donor EF-G (bifunctional rhodamine, residues 467–474), acceptor uL11 (Cy5, residue 87)[73]. **p,** Schematic of translocation indicating the positions of donor (LD550, uS13 N terminus) and acceptor (LD650, uL1) fluorophores and FRET efficiency values for the indicated states in **q–t. q–t,** Population FRET histograms showing time evolution of FRET between uS13 and uL1 upon injection of 5 μM EF-G with either 1 mM GTP (**q**, Apo), 1 mM GTP and 3 mM SPC (**r**), 1 mM GTPγS (**s**) or 1 mM GTP and 400 μM fusidic acid (**t**, FA), revealing that the non-hydrolysable GTP analogue GTPγS stalls the ribosome in both the same states as SPC (INT1) and fusidic acid (INT2), whereas SPC stalls only in INT1 and fusidic acid stalls only in INT2. *N* indicates the number of observed molecules.

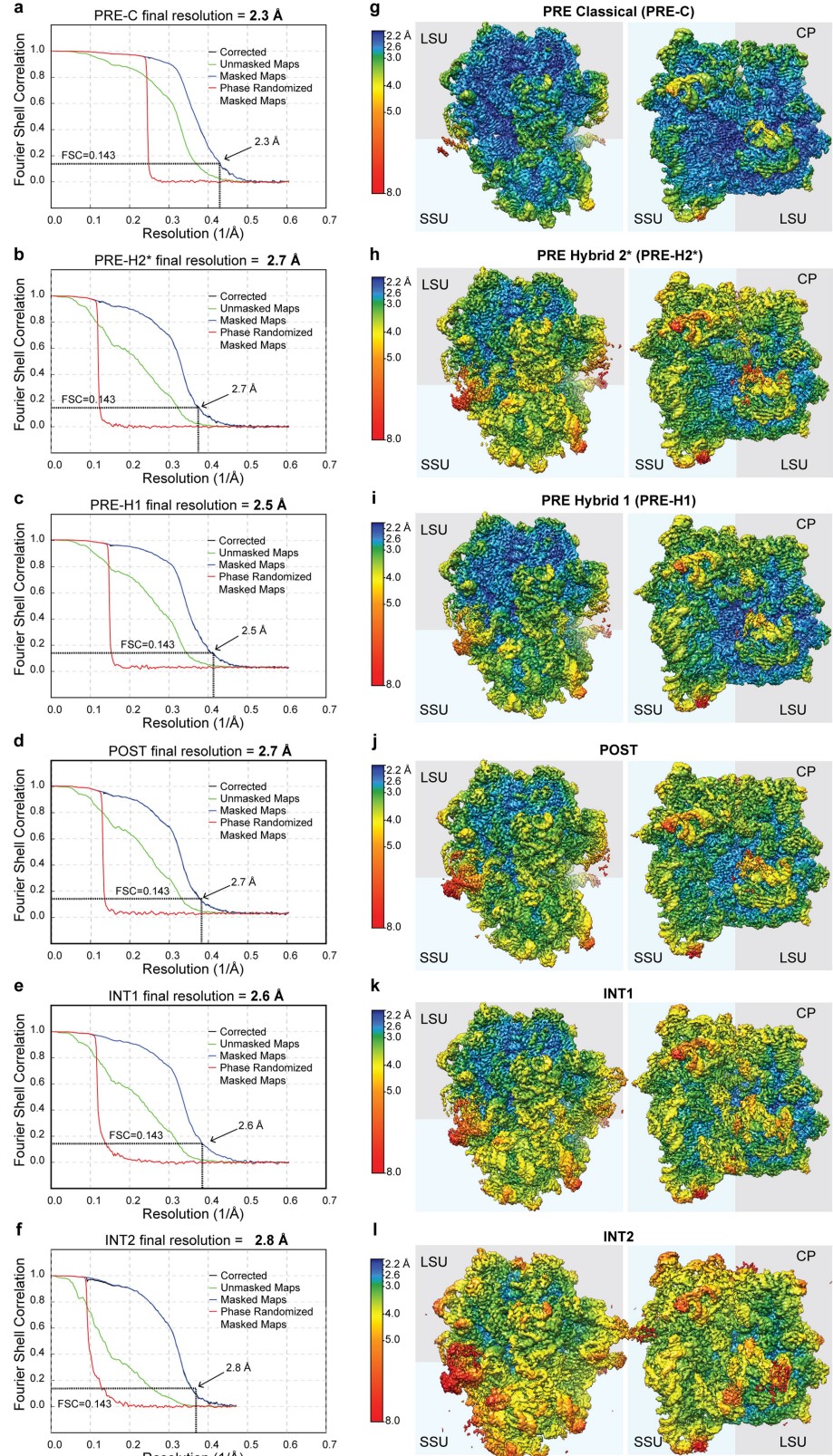

**Extended Data Fig. 2 | Fourier shell correlation and local resolution for cryo-EM structures along the translocation reaction coordinate.** **a**–**f**, Fourier shell correlation (FSC) resolution curves for PRE-C (**a**), PRE-H2* (**b**), PRE-H1 (**c**), POST (**d**), INT1 (**e**) and INT2 (**f**) structures obtained by masking the two half maps and calculating the cross-resolution between the masked

volumes in Relion. Resolution was estimated using the 0.143 cut-off criterion (dotted line). **g**–**l**, Local resolution electron density maps for PRE-C (**g**), PRE-H2* (**h**), PRE-H1 (**i**), POST (**j**), INT1 (**k**) and INT2 (**l**). CP, central protuberance. LSU, grey background; SSU, blue background. Threshold $\sigma = 5$. See also Methods and Supplementary Table 1.

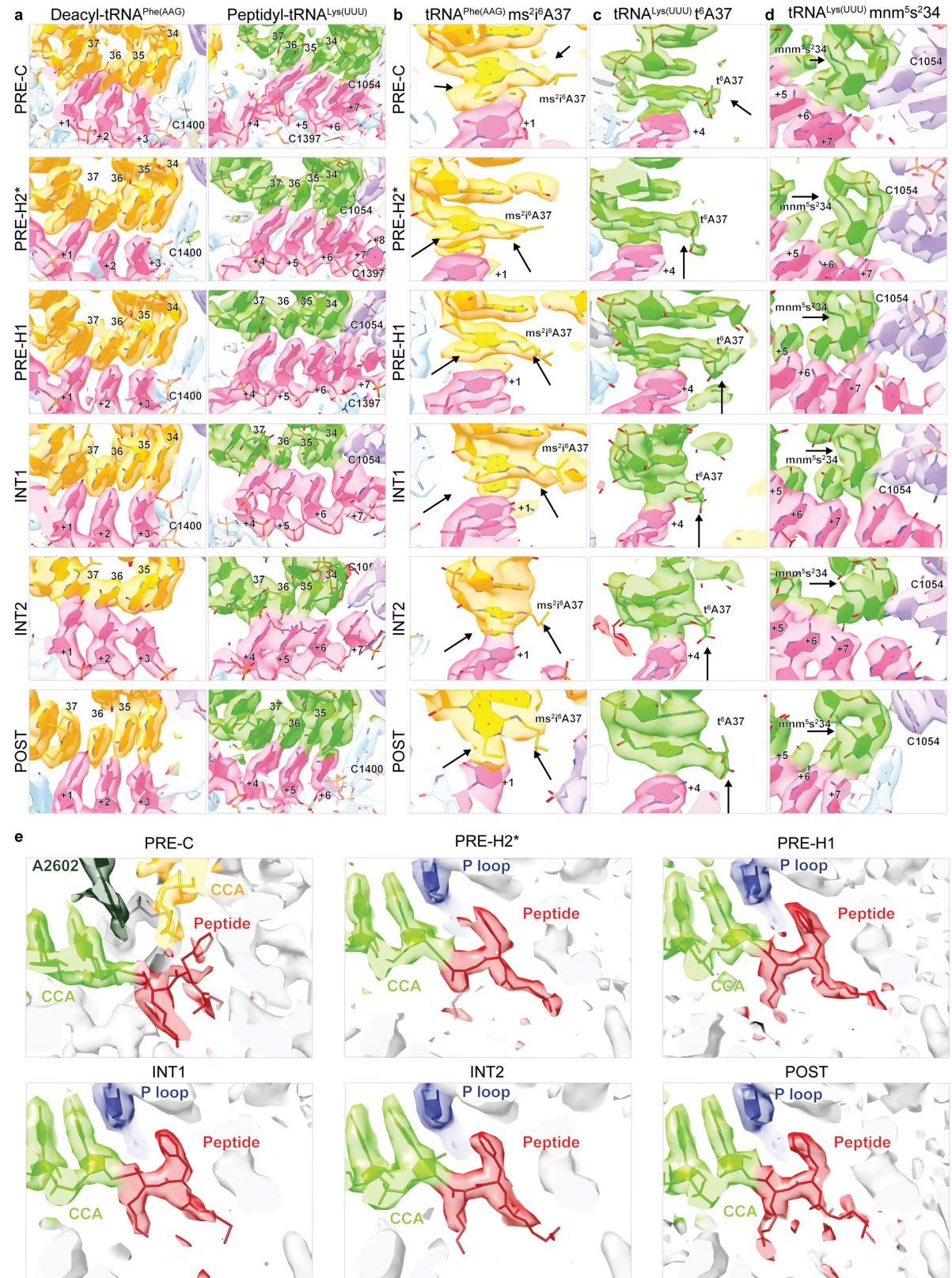

**Extended Data Fig. 3 | Structural evidence of tRNA identity for cryo-EM structures along the translocation reaction coordinate. a**, Locally filtered electron density for cognate tRNA anticodon–mRNA codon (pink) interactions with deacyl-tRNA^Phe (AAG, left, yellow/orange) and peptidyl-tRNA^Lys (UUU, right, green) from PRE-C (top) to POST (bottom). SSU head (purple) and body (blue). Threshold $\sigma = 6$. **b–d**, Locally filtered electron density for modified tRNA bases in support of tRNA assignment from PRE-C (top) to POST (bottom): deacyl-tRNA^Phe(AAG) ms²i⁶A37 (MIA, **b**); peptidyl-tRNA^Lys(UUU) t⁶A37 (T6A, **c**); peptidyl-tRNA^Lys(UUU) mnm⁵s²34 (U8U, **d**). Arrows designate defining density for each modification. Coloured as in **a**. **e**, Locally filtered electron density for the nascent peptide (fMet-Phe-Lys, red). Deacyl-tRNA, yellow/orange; peptidyl-tRNA, green; A2602, dark green; P loop, dark blue. Threshold $\sigma = 5$ for PRE-C, INT1 and INT2. Threshold $\sigma = 7$ for PRE-H2*, PRE-H1 and POST.

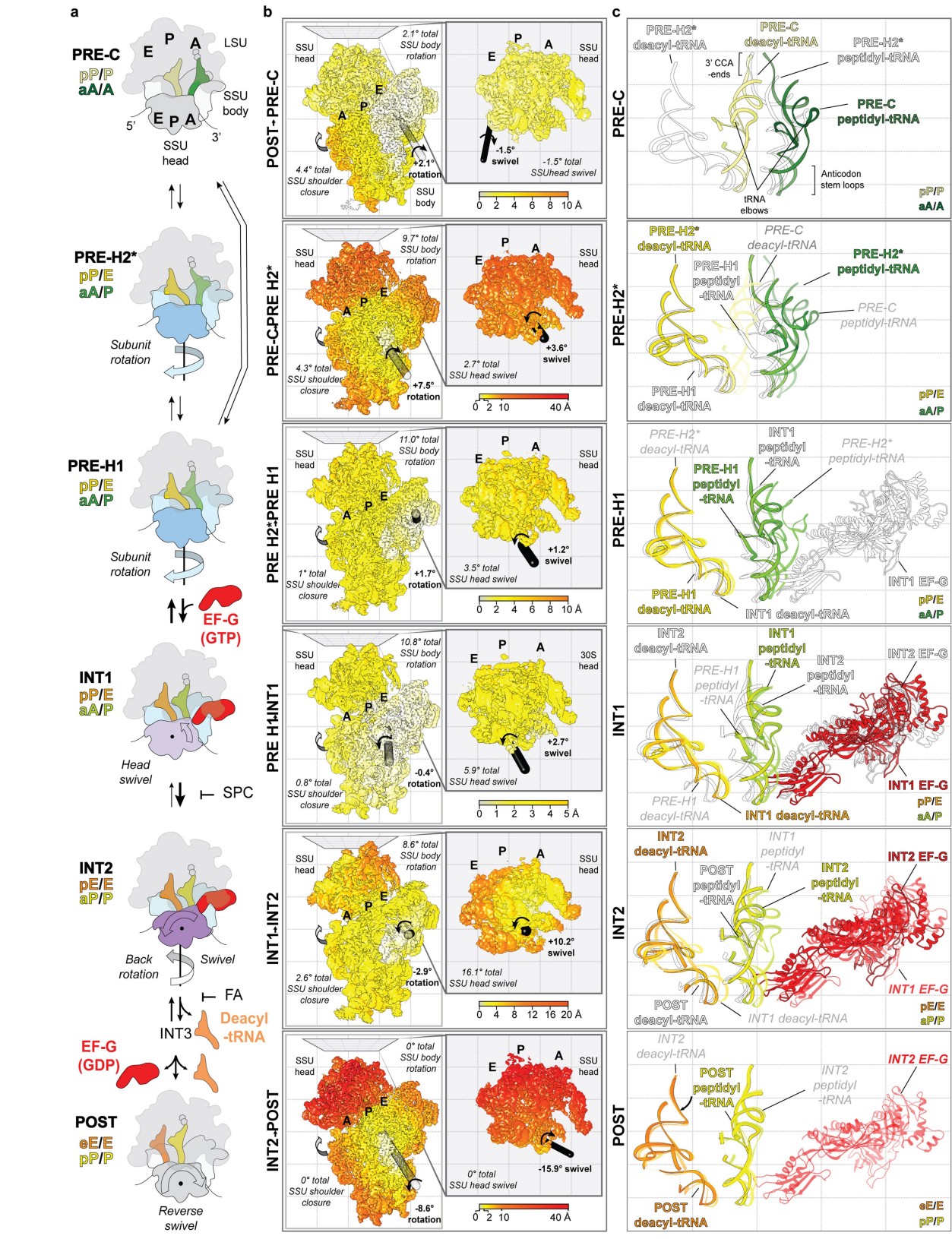

**Extended Data Fig. 4** | See next page for caption.

**Extended Data Fig. 4 | Global conformational changes within the ribosome that define the translocation reaction coordinate. a**, Schematic of the translocation reaction coordinate in bacteria depicting SSU rotation (blue) with respect to the LSU (grey) and SSU head-swivel (purple) processes. tRNAs are coloured on a gradient from the A (green) to P (yellow) to E (orange) sites. Deacyl-tRNA dissociation (orange) can occur at multiple steps after INT2. tRNA positions are depicted in chimeric-hybrid notation (ssu head SSU BODY/LSU). **b**, SSU conformational changes accompanying each sequential translocation step, viewed from inside the intersubunit space (left) and towards the intersubunit space from the head domain (inset), coloured by r.m.s.d. at each SSU residue for each transition. Degree of shoulder domain closure, SSU body-rotation and SSU head-swivel as compared to POST is indicated as 'total'. Cylindrical axes and bolded text indicate the degree of SSU body-rotation (transparent black axis, right, LSU core alignment) and SSU head-swivel (solid black axis, right, SSU body alignment) as compared to the previous state on the reaction coordinate. Threshold $\sigma = 5$. **c**, Deacyl-tRNA (yellow/orange), peptidyl-tRNA (green/yellow) and EF-G (red) movements during translocation. Current tRNA and EF-G positioning (solid coloured, outlined), previous position (transparent colour, no outline) and next position (white, solid outline). Alignment on the LSU core. Camera perspective is identical for all images. See also Supplementary Table 2.

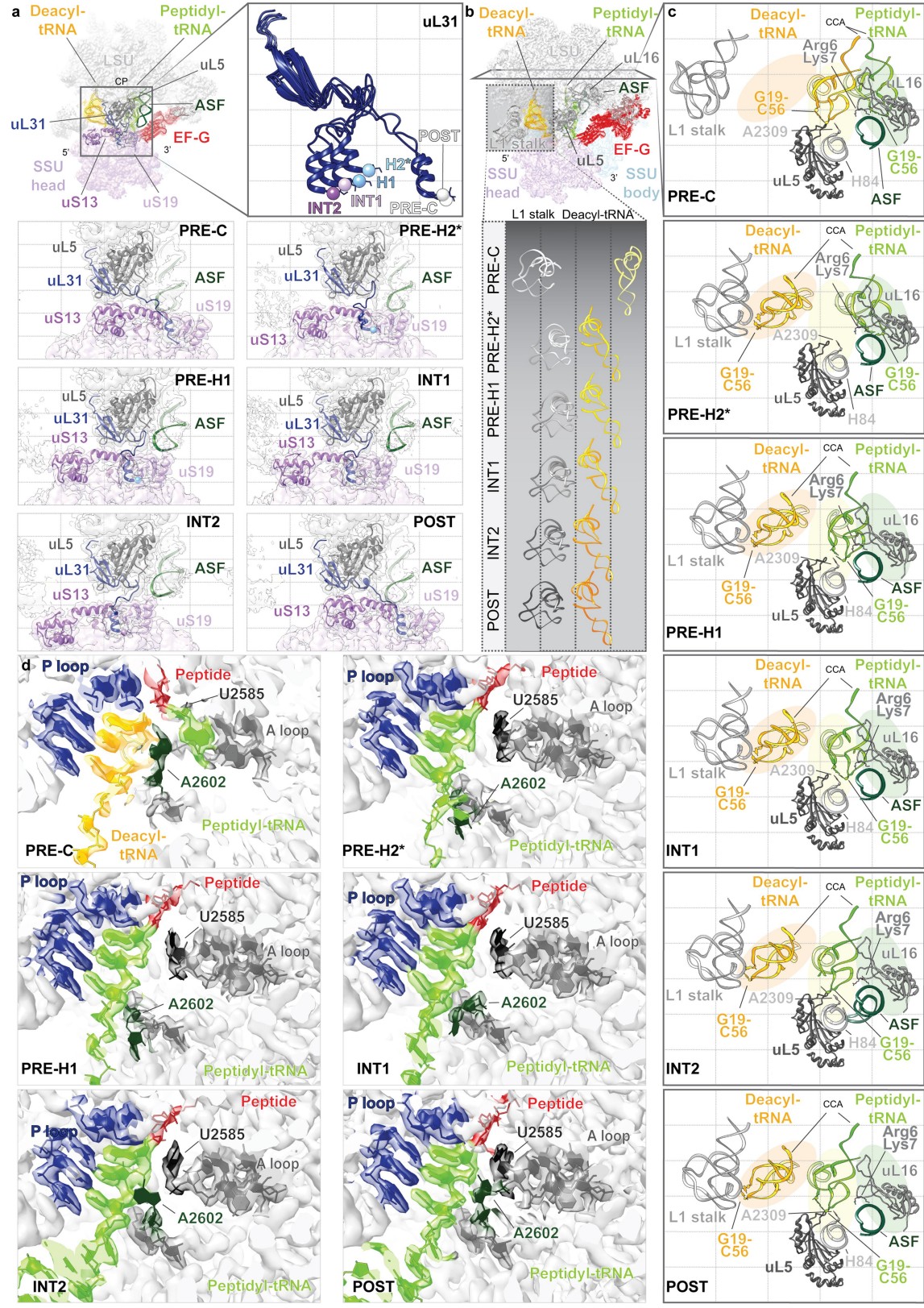

**Extended Data Fig. 5** | See next page for caption.

**Extended Data Fig. 5 | LSU interactions with tRNA and the SSU in each characterized ribosome structure along the translocation reaction coordinate. a**, Location of intersubunit bridge B1 on the ribosome (top, left) and overlay of uL31 model (dark blue) depicting the conformational change during translocation (top, right). Sphere is positioned on residue Ile66 (Cα). Bottom, bridge B1 interactions between the LSU central protuberance and the SSU head domain (purple) in each structural state, including the ASF (H38, dark green) to uS13 (B1a), uL5 to uS13 (B1b) and uL31 to uS13/uS19 (B1c). Threshold $\sigma = 4$. **b**, Location of the L1 stalk, uL5 (dark grey), uL16 (light grey) and ASF (dark green) on the INT1 ribosome (top). L1 stalk positioning and interaction with deacyl-tRNA (yellow/orange) elbow in each classified structure (bottom). **c**, Bridge B1 interactions between the central protuberance and deacyl-/ peptidyl-tRNA from PRE-C (top) to POST (bottom). P-site tRNA G19–C56 pair contacts include LSU H84 base A2309 and uL5 N-terminal Arg80. A-site tRNA contacts include G19–C56 pair coordination by ASF base A896 and TΨC stem packing against uL16 residues Arg6 and Lys7. A site (green circle), P site (yellow circle), E site (orange circle) regions are shown. **d**, Locally filtered electron density illustrating peptidyl transferase centre interactions along the translocation interaction coordinate. Watson–Crick pairing between the 3′-CCA tRNA ends and the peptidyl transferase centre. P loop (LSU rRNA 2251–2253, 2450–2451), dark blue; A loop (LSU rRNA 2553–2555, 2582–3585, dark grey; peptide, red; LSU base A2602, dark green; deacyl-tRNA, yellow-orange; peptidyl-tRNA, green; Threshold $\sigma = 7$. Camera perspective is identical for images within each panel. Alignment on the LSU core. See also Supplementary Video 2.

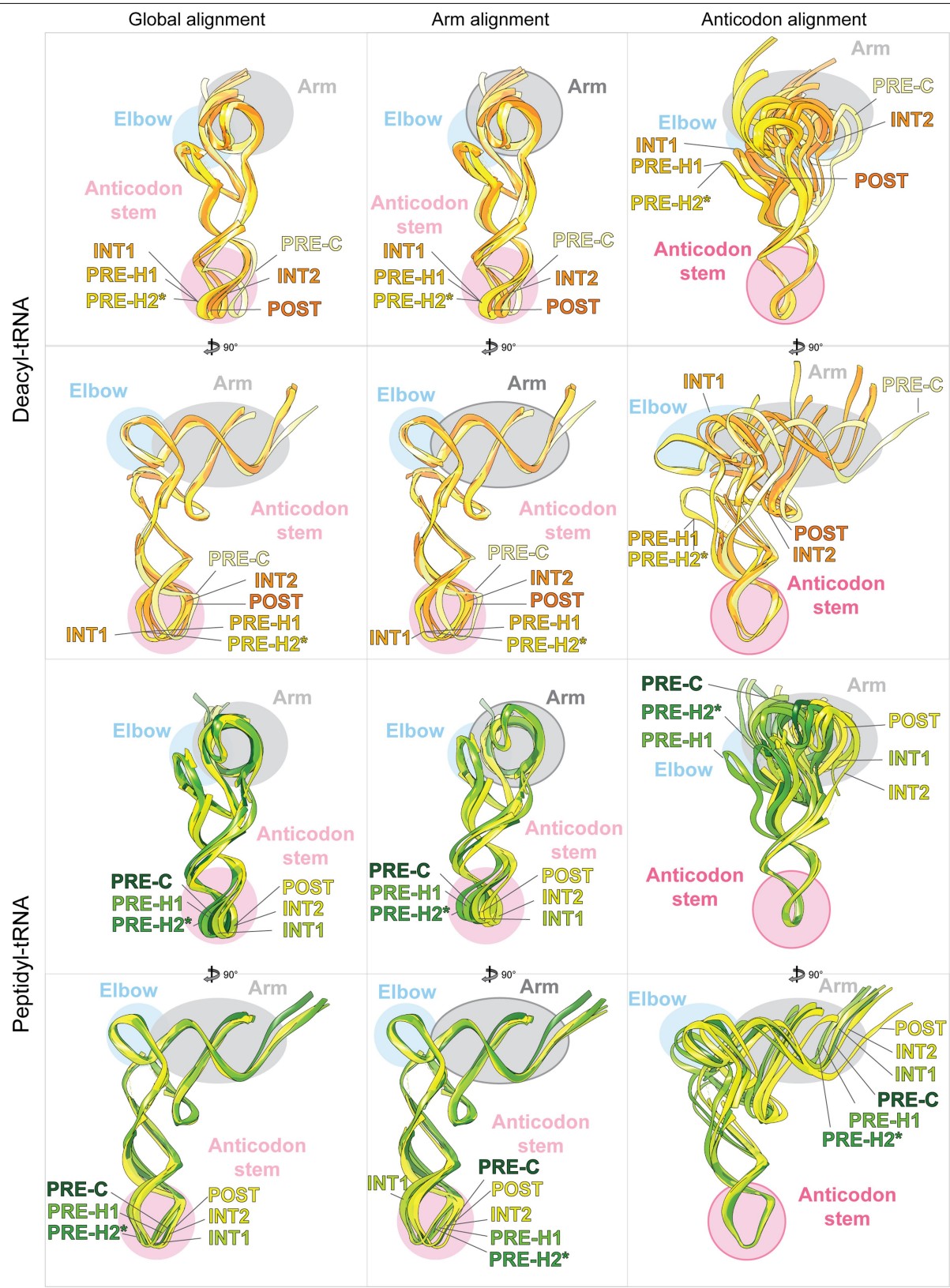

**Extended Data Fig. 6 | Changes in tRNA conformation during translocation.** Alignment of deacyl-tRNA (rows 1 and 2, orange) and peptidyl-tRNA (bottom, green) globally (left column), on the tRNA acceptor arm domain (positions 1–6 and 50–72, middle column, grey oval) or on the tRNA anticodon stem loop (positions 30–40, right column, pink circle). Deacyl-tRNA is coloured on a gradient from yellow to orange based on the position on the reaction coordinate (PRE-C, PRE-H2*, PRE-H1, INT1, INT2 and POST). Peptidyl-tRNA is coloured on a gradient from green to yellow. Blue circle annotates the position of the tRNA elbow. Alignment is on the outlined circle. See also Supplementary Table 2 and Video 4.

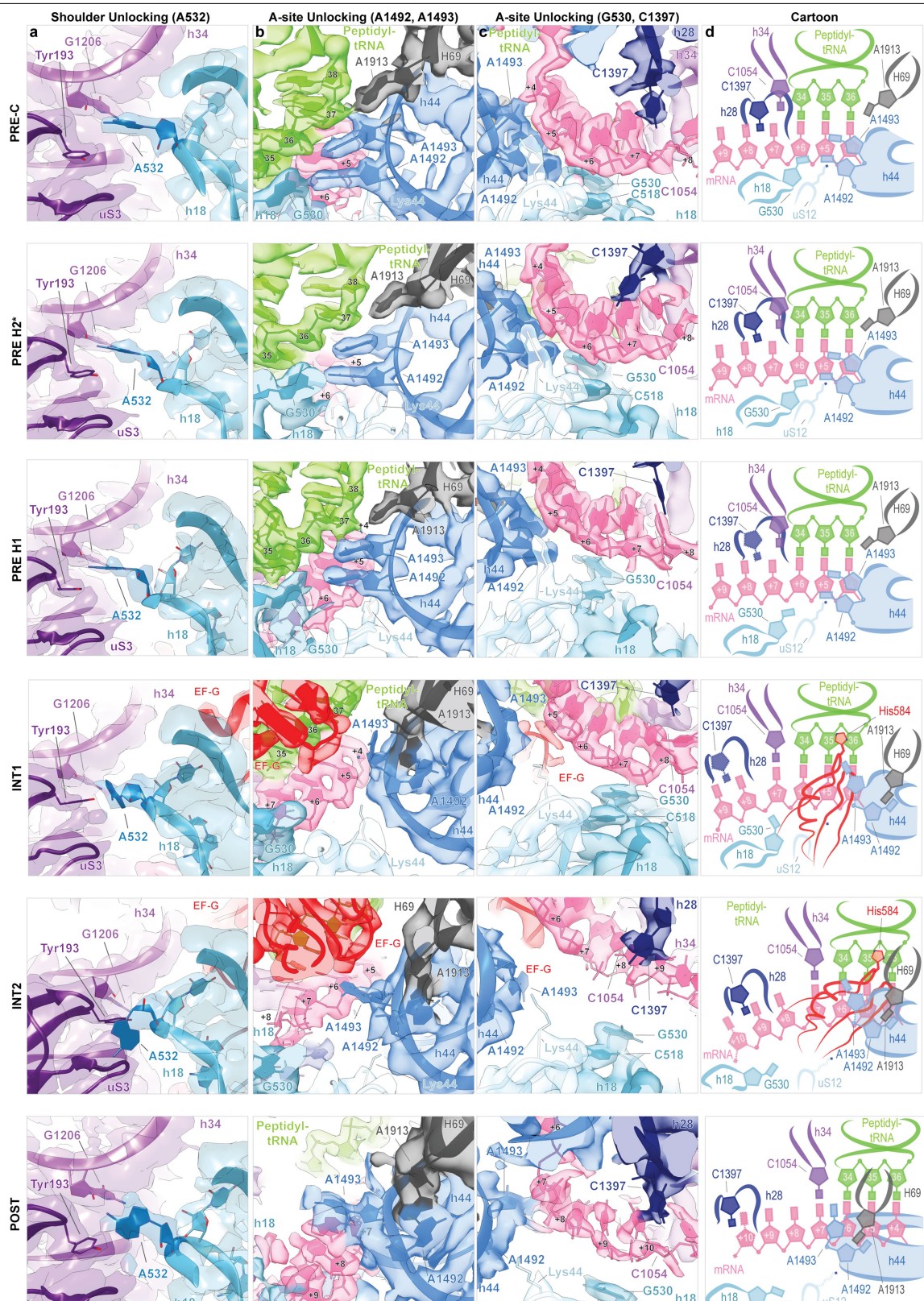

**Extended Data Fig. 7 | SSU unlocking on the leading-edge during translocation. a**, Locally filtered electron density illustrating the interaction between the SSU shoulder domain (cyan) A532 base, the SSU head domain (purple) G1206 of h34 and uS3 (dark purple) 193 loop near the A-site mRNA and EF-G (red) binding site. **b**, Locally filtered electron density illustrating the points of contact involved in unlocking the tRNA₂–mRNA module from the SSU body domain monitoring bases (A1492, A1493) in the A site. Peptidyl-tRNA,

green; mRNA, pink; H69, grey; h44, blue; h28, dark blue; h18, cyan; uS12, light blue; h34, purple; EF-G, red. **c**, Locally filtered electron density and molecular models from **b**, viewed from beneath the SSU A site. Coloured as in **b**. **d**, Cartoon schematic depicting A-site unlocking. Dotted lines indicate weak electron density. Coloured as in **b**, **c**. Camera perspective is identical for images in each panel. Alignment on LSU core. Threshold $\sigma$ = 5. See also Fig. 2.

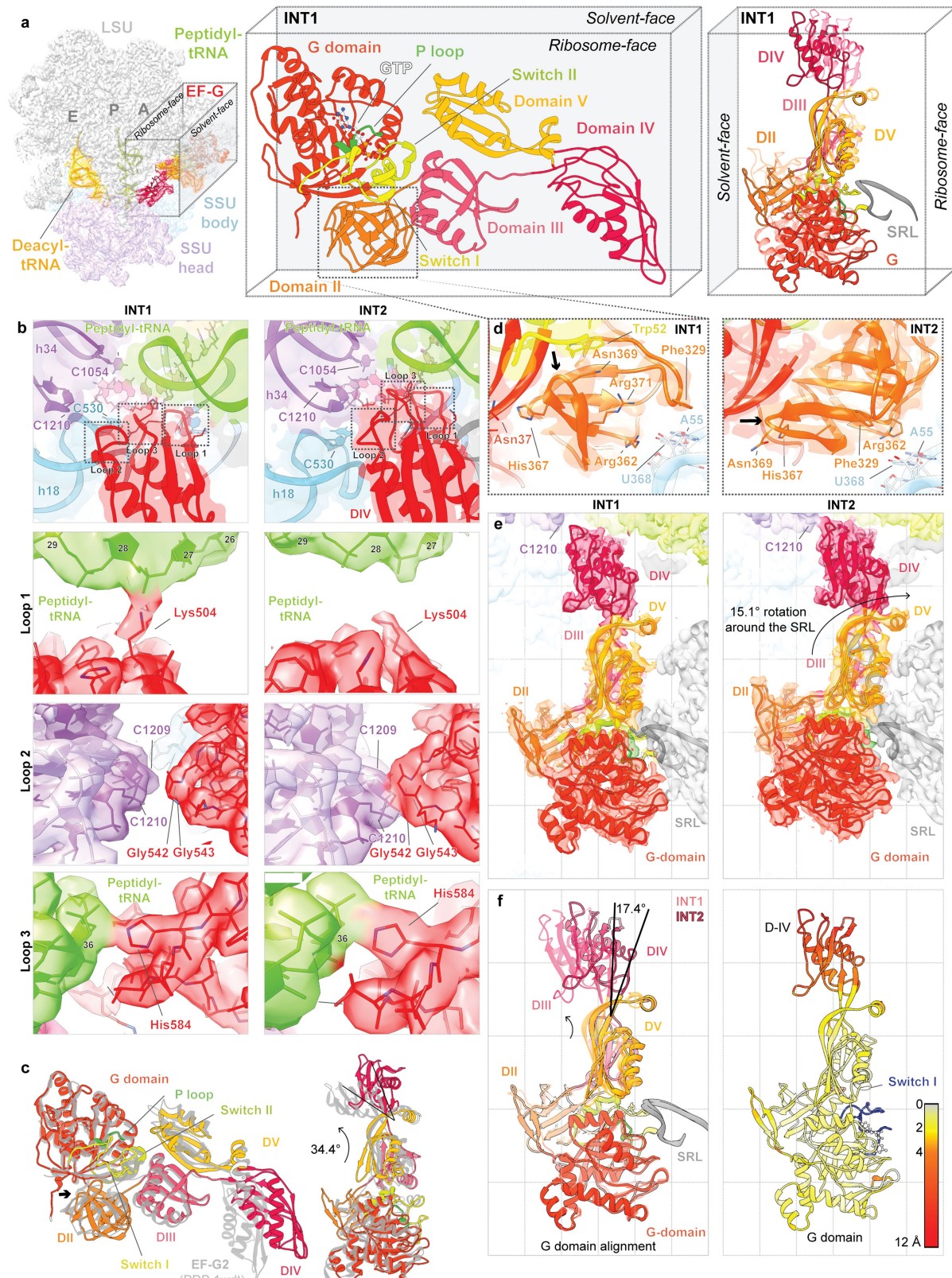

**Extended Data Fig. 8** | See next page for caption.

**Extended Data Fig. 8 | Overview of the EF-G conformations and interactions in INT1 and INT2. a**, EF-G binding site (left) and domain architecture (middle and right) in its active, GTP conformation (INT1). Model for INT2 EF-G is shown as transparent in the right panel. **b**, Interaction between EF-G DIV (red) and the mRNA (pink) peptidyl-tRNA (green) codon–anticodon minihelix in the A site in the INT1 (left) and INT2 (right) structures (top). Locally filtered electron density illustrating the interaction between EF-G DIV loop I Lys504 and peptidyl-tRNA (row 2), EF-G DIV loop II Gly542 and C1210 of SSU h34 (purple, row 3) and EF-G DIV loop II His584 and peptidyl-tRNA (row 4). Threshold $\sigma = 3$. **c**, GTP/GDP-$P_i$ conformation of EF-G (INT1, coloured) compared to a published structure of an EF-G homologue (grey, EF-G-2, PDB ID: 1WDT[37]). Degree of bend angle between the G domain and DIV compared to INT1 is indicated. Conformational change in DII is indicated with an arrow. Coloured as in **a**. **d**,

EF-G DII conformational change (arrow) in the INT1 (left) to INT2 (right) transition, oriented by the G domain (Asn37), switch I (Trp52) and DII (His367, Asn369 and Arg371), rearranging DII contact with SSU h5 via Phe329 and Arg362. Coloured as in **a**. **e**, EF-G rotation of approximately 15° around the SRL (grey) and into the A site in the INT1 (left) to INT2 (right) transition. This movement placed SSU head nucleotide C1210 (h34, purple) in direct contact with EF-G DIV in INT2. Coloured as in **a**. Threshold $\sigma = 5$. **f**, Conformational rotation of the superdomain DI–DIII of EF-G in the INT1 (solid) to INT2 (transparent) transition (approximately 17°, left). Models of EF-G were superimposed on the G domain. r.m.s.d. of EF-G in the INT1 to INT2 transition mapped on the INT1 structure (right). Density absent in the INT2 structure is depicted in dark blue. Alignment on the LSU core, unless otherwise stated.

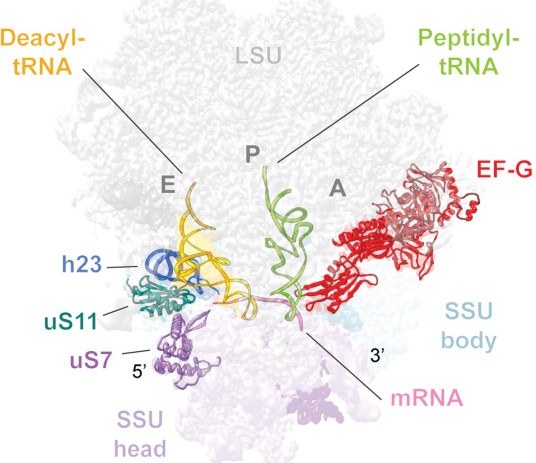

**Extended Data Fig. 9 | E-site mRNA codon interactions with SSU h23, uS7 and uS11.** Locally filtered electron density illustrating the interaction between mRNA and the SSU in the E site. The black arrow annotates the −1 mRNA position, which flips in base orientation during translocation. The orange arrow annotates the gap between SSU proteins uS7 and uS11. mRNA, pink; uS7, purple; uS11, cyan; h23, blue; h24, light blue; h28, dark blue; deacyl-tRNA, yellow; peptidyl-tRNA, green. A putrescine molecule (PUT, grey) is modelled proximal to the −1 ribose in the PRE-H2* and PRE-H1 states. Alignment on the LSU core. Threshold $\sigma = 5$.

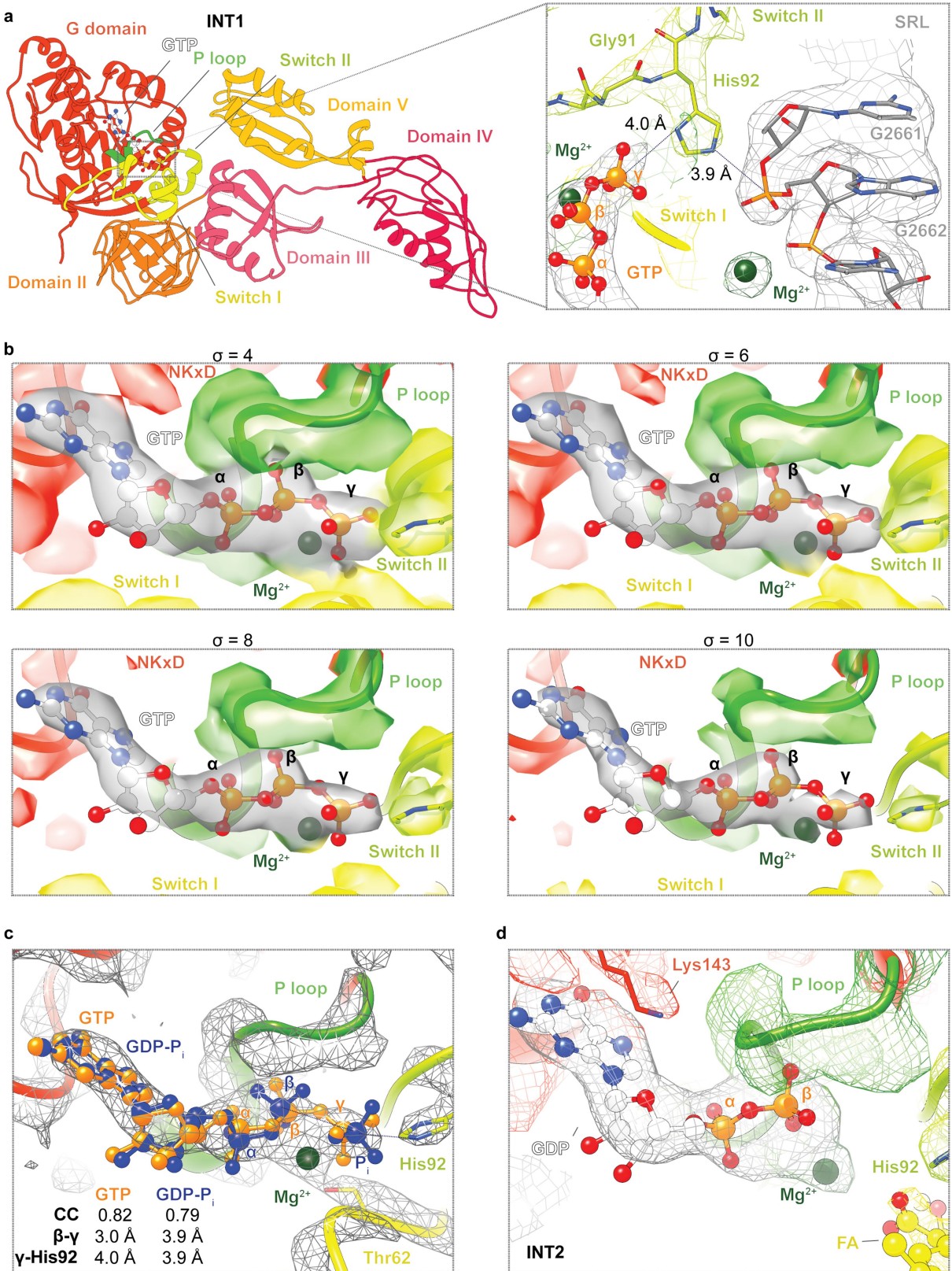

**Extended Data Fig. 10 | EF-G nucleotide-binding pocket in INT1 and INT2.**
**a**, INT1 EF-G domain architecture (left) and zoom in of locally filtered electron density illustrating nucleotide interactions, including catalytic residues His92 of switch II and G2661 and G2662 of the SRL (grey, right). Threshold $\sigma = 6$.
**b**, Locally filtered electron density at different thresholds ($\sigma = 4, 6, 8, 10$) for the phosphates in the nucleotide-binding pocket of EF-G in INT1. **c**, Alternative modelling of GDP-P$_i$ (dark blue) in the nucleotide-binding pocket of INT1 compared to GTP (orange). Ligand cross correlation (CC) value reported from the corresponding PDB validation reports. Distances were measured between $\gamma$ phosphate, $\beta$ phosphate and His92 amine group. **d**, Locally filtered electron density illustrating the nucleotide-binding pocket of EF-G in the INT2 structure. Threshold $\sigma = 6$. G domain, red; DII, dark orange; DIII, strawberry; DIV, hot pink; DV, yellow/orange; P loop, green; switch I, yellow; switch II, lime; Mg$^{2+}$, dark green. See also Fig. 3.

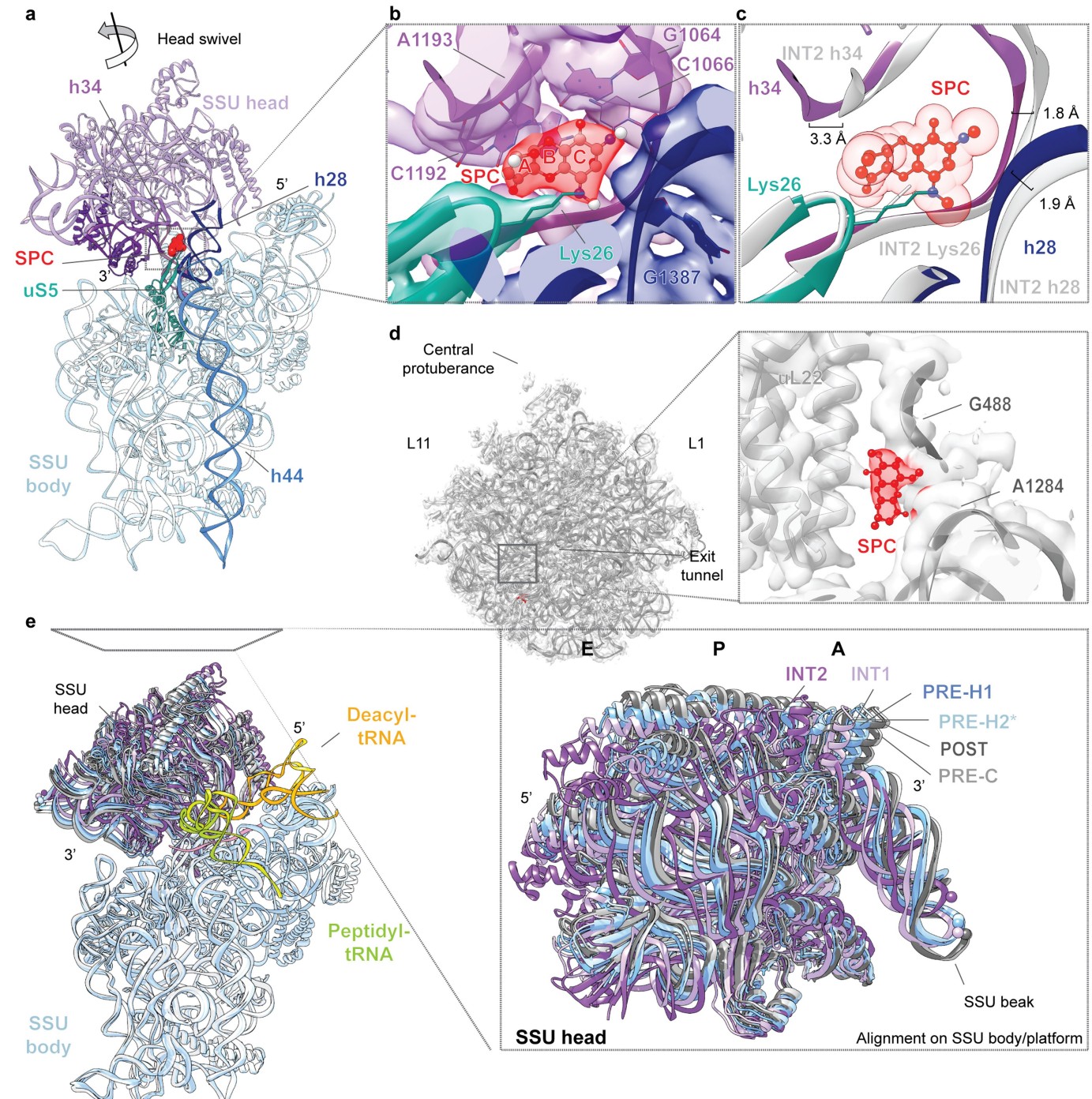

**Extended Data Fig. 11 | SPC-binding sites evidenced in INT1. a**, SPC-binding site (red) on the SSU between the head (purple) and body (blue) domains beneath the SSU P site. **b**, Zoom-in of the primary SPC-binding site showing that the convex face of SPC sits in the major groove of h34 (purple, between A1193 and G1064) and that the concave face of SPC sits between h28 (dark blue, G1387) and uS5 (cyan). Lys26 of uS5 reaches to interact with the h28 phosphate G1387. **c**, Overlay of the SSU SPC-binding pocket for INT1 (coloured) and INT2 (white), showing collapse of the SPC binding site and reorientation of Lys26 during SSU head-domain swivel. SPC shown in ball-and-stick and transparent-sphere representation. Distance changes for h28 (G1387 C1'), h34 (A1196 C1') and h35 (C1066 C1') are indicated. **d**, Electron density indicative of a second SPC-binding site near the exit tunnel of the LSU between uL22 and nucleotides G488 and A1284. Threshold $\sigma = 6$. **e**, Overlay of the SSU from PRE-C (light grey), PRE-H2* (light blue), PRE-H1 (blue), INT1 (light purple), INT2 (purple) and POST (dark grey) aligned on the SSU body domain to illustrate head-domain swivel. See also Supplementary Video 6.

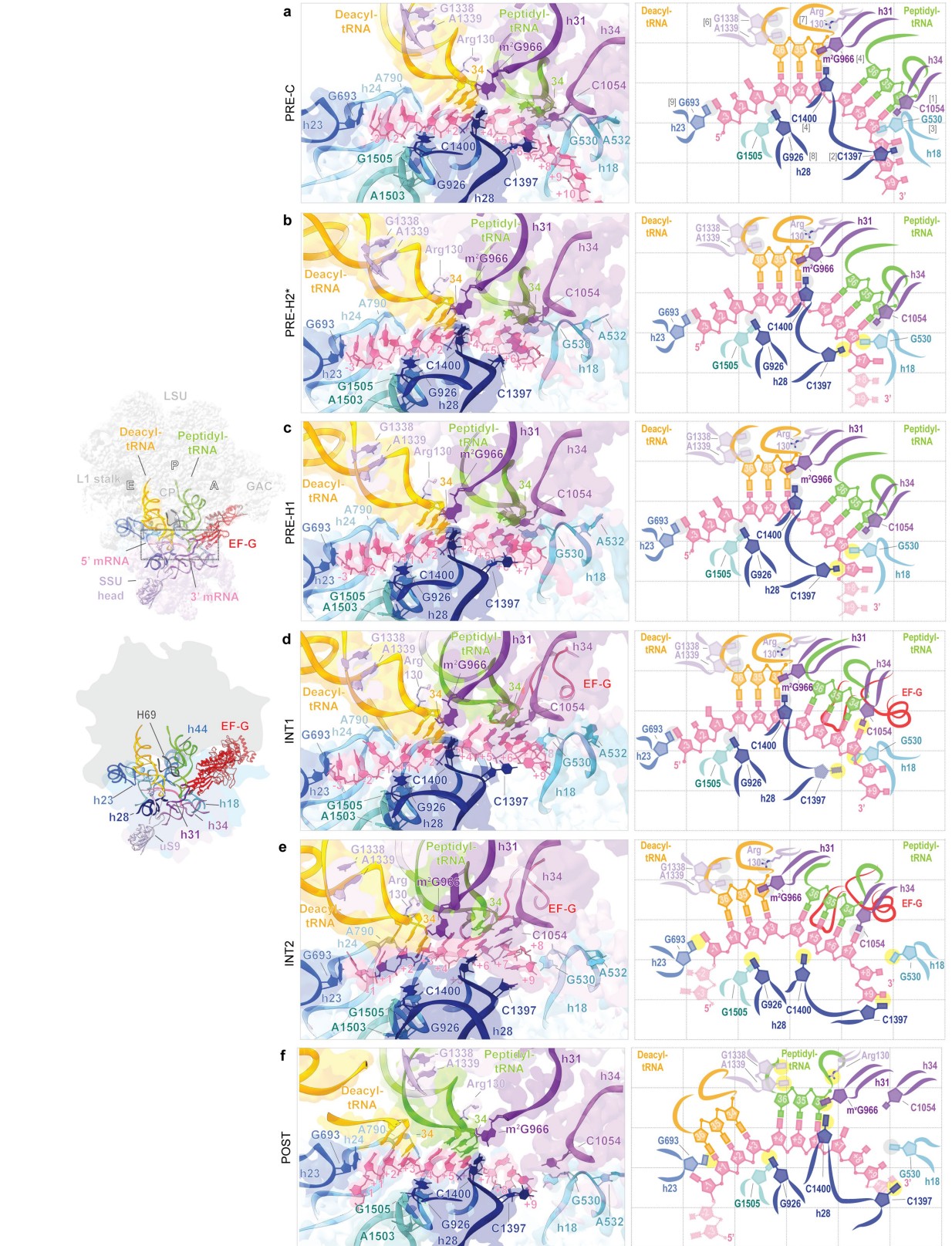

**Extended Data Fig. 12** | See next page for caption.

**Extended Data Fig. 12 | Coordinated translocation of the tRNA$_2$–mRNA module.** Overview of the helical architecture involved in tRNA$_2$–mRNA translocation (left). **a**–**f**, Interactions between the SSU and the tRNA$_2$–mRNA module depicted with molecular model and electron density (left) together with a schematic representation (right) for PRE-C (**a**), PRE-H2* (**b**), PRE-H1 (**c**), INT1 (**d**), INT2 (**e**) and POST (**f**). A-site contacts with the tRNA$_2$–mRNA module depicted here include (1) anchored h34 (plum) SSU head base C1054 stacking on tRNA at position 34, (2) intercalation of h28 (dark blue) C1397 into mRNA (pink) downstream of the A-site codon (PRE +7/+8, POST +10/+11) and (3) hydrogen bonding of the h18 (cyan) SSU shoulder G530 base with the A-site wobble position. P-site interactions include (4) anchored stacking of h31 (purple) SSU head base m$^2$G966 at tRNA position 34, (5) h28 C1400 base stacking against the P-site wobble position, (6) A-minor interactions of SSU head bases G1338/A1339 with the tRNA minor groove, (7) electrostatic interactions of uS9 (light purple) C-terminal Arg130 residue with the anticodon U-turn motif and (8) hydrogen bonding of the h28 G926 base with the phosphate backbone between the −1 and +1 mRNA positions, anchored by A1505 (aqua). E-site interactions include (9) the anchored stacking of the SSU body h23 (light blue) G693 base against the −3 mRNA base. Distances in the schematic are not to scale. Dotted residues display weak electron density. Grey circles depict contacts that are unchanged from the previous state, yellow circles depict contacts that are different from the previous state. Peptidyl-tRNA, green; deacyl-tRNA, orange. Camera perspective is identical for all images. Alignment on the LSU core. Threshold $\sigma = 6$.

# nature research

# Reporting Summary

Nature Research wishes to improve the reproducibility of the work that we publish. This form provides structure for consistency and transparency in reporting. For further information on Nature Research policies, see our Editorial Policies and the Editorial Policy Checklist.

## Statistics

For all statistical analyses, confirm that the following items are present in the figure legend, table legend, main text, or Methods section.

| n/a | Confirmed | |
|---|---|---|
| ☐ | ☒ | The exact sample size (*n*) for each experimental group/condition, given as a discrete number and unit of measurement |
| ☐ | ☒ | A statement on whether measurements were taken from distinct samples or whether the same sample was measured repeatedly |
| ☒ | ☐ | The statistical test(s) used AND whether they are one- or two-sided *Only common tests should be described solely by name; describe more complex techniques in the Methods section.* |
| ☒ | ☐ | A description of all covariates tested |
| ☒ | ☐ | A description of any assumptions or corrections, such as tests of normality and adjustment for multiple comparisons |
| ☐ | ☒ | A full description of the statistical parameters including central tendency (e.g. means) or other basic estimates (e.g. regression coefficient) AND variation (e.g. standard deviation) or associated estimates of uncertainty (e.g. confidence intervals) |
| ☒ | ☐ | For null hypothesis testing, the test statistic (e.g. *F*, *t*, *r*) with confidence intervals, effect sizes, degrees of freedom and *P* value noted *Give P values as exact values whenever suitable.* |
| ☒ | ☐ | For Bayesian analysis, information on the choice of priors and Markov chain Monte Carlo settings |
| ☒ | ☐ | For hierarchical and complex designs, identification of the appropriate level for tests and full reporting of outcomes |
| ☒ | ☐ | Estimates of effect sizes (e.g. Cohen's *d*, Pearson's *r*), indicating how they were calculated |

*Our web collection on statistics for biologists contains articles on many of the points above.*

## Software and code

Policy information about availability of computer code

| Data collection | The time-evolution of FRET signal was then recorded using a home-built total internal reflection based fluorescence microscope at ~0.1 kW/cm2 laser (532 nm) illumination at a time resolution of 40 ms or 400 ms. Donor and acceptor fluorescence intensities were extracted from the recorded movies and FRET efficiency traces were calculated using custom software implemented in MatlabR2015b.<br><br>Cryo-EM data collection was performed using SerialEM software (version 3.7.1) with image shift protocol (9 images were collected with one defocus measurements per 9 holes). |
|---|---|
| Data analysis | smFRET traces were analyzed using hidden Markov model idealization methods as implemented in the SPARTAN software package (version 3.7.0).<br><br>Motion correction of cryo-EM micrographs was performed on raw super resolution movie stacks and binned 2-fold using MotionCor2 software. CTF parameters were determined using CTFFind4 and refined later in Relion 3.1 and cryoSPARC 3. Prior to particle picking, good micrographs were qualified by power spectrum. Particles were picked within cisTEM 1 and the coordinates were transferred to Relion 3.1. |

For manuscripts utilizing custom algorithms or software that are central to the research but not yet described in published literature, software must be made available to editors and reviewers. We strongly encourage code deposition in a community repository (e.g. GitHub). See the Nature Research guidelines for submitting code & software for further information.

## Data

Policy information about availability of data

All manuscripts must include a data availability statement. This statement should provide the following information, where applicable:

- Accession codes, unique identifiers, or web links for publicly available datasets
- A list of figures that have associated raw data
- A description of any restrictions on data availability

PDBs and cryo-EM 3D maps for all structures are available through the Protein Data Bank (PDB, www.rcsb.org) and Electron Microscopy Data Bank (EMDB, www.ebi.ac.uk/pdbe/emdb/), respectively (PRE-C, PDB-ID 7N1P, EMD-24120; PRE-H2*, PDB-ID 7N30, EMD-24135; PRE-H1, PDB ID 7N2U, EMD-24133; INT1-SPC, PDB-ID 7N2V, EMD-24134; INT2-FA, PDB-ID 7N2C, EMD-24132; POST, PDB-ID 7N31, EMD-24136). The PDBs that were used as templates for model building (PDB-IDs 4ybb, 5e81, 4wro, 4v9o, 1ctf) are available at www.rcsb.org.

# Field-specific reporting

Please select the one below that is the best fit for your research. If you are not sure, read the appropriate sections before making your selection.

☒ Life sciences ☐ Behavioural & social sciences ☐ Ecological, evolutionary & environmental sciences

For a reference copy of the document with all sections, see nature.com/documents/nr-reporting-summary-flat.pdf

# Life sciences study design

All studies must disclose on these points even when the disclosure is negative.

| | |
|---|---|
| Sample size | No statistical methods were used to predetermine sample size. All quantitative experiments were carried out in triplicate and means and standard deviations were calculated. All replicates were included in the analysis. The number of micrographs collected was determined by the number of particles required to achieve a high resolution reconstruction of the target complex. |
| Data exclusions | No data was excluded from smFRET experiments. Cryo-EM data analysis and classification can be found in Supplementary Information. |
| Replication | All quantitative measurements were carried out in triplicate and all repeats were included in the analysis, when ap. Cryo-EM data was processed according to standard methods and multiple datasets were collected yielding the same results. |
| Randomization | The experiments were not randomized as it is not applicable to structural determination and dynamics studies. |
| Blinding | The investigators were not blinded to allocation during experiments and outcome assessment as none of the experiments involved either human or animal models or group allocation. |

# Reporting for specific materials, systems and methods

We require information from authors about some types of materials, experimental systems and methods used in many studies. Here, indicate whether each material, system or method listed is relevant to your study. If you are not sure if a list item applies to your research, read the appropriate section before selecting a response.

| Materials & experimental systems | | Methods | |
|---|---|---|---|
| **n/a** | **Involved in the study** | **n/a** | **Involved in the study** |
| ☒ ☐ | Antibodies | ☒ ☐ | ChIP-seq |
| ☒ ☐ | Eukaryotic cell lines | ☒ ☐ | Flow cytometry |
| ☒ ☐ | Palaeontology and archaeology | ☒ ☐ | MRI-based neuroimaging |
| ☒ ☐ | Animals and other organisms | | |
| ☒ ☐ | Human research participants | | |
| ☒ ☐ | Clinical data | | |
| ☒ ☐ | Dual use research of concern | | |

