## [Peer Review File · Nature]

Manuscript Title: Structural basis of early translocation events on the ribosome

Reviewer Comments & Author Rebuttals

Reviewer Reports on the Initial Version:

Referees' comments:

Referee #1:

The manuscript of Rundlett et al. represents a major step forward in understanding the mystery of ribosomal translocation. Translocation involved the coordinated movement of the P- and A-site tRNAs after peptide bond formation that is catalyzed by a conserved GTPase, EF-G. Decades of biochemical, structural and dynamics work have outlined the global steps of translocation. Upon peptide bond formation, the small and large subunits rotate with respect to each other and tRNAs move into a dynamic hybrid conformation; it is this state that is recognized by EF-G. Translocation is rapid, accelerated at least 50-fold by GTP hydrolysis, occurring within 50 ms. Despite this, structures of many of these intermediate conformations have remained unsolved, and how GTP energy is used during this directional movement is not clear. Here, guided by the elegant and rigorous single-molecule FRET studies of Blanchard group, the authors identified buffer conditions plus antibiotics that greatly slowed translocation, allowing the design of cryoEM experiments that populate the previously unobserved intermediates. It is a wonderful mixture of dynamics and cryoEM, and builds on the more than decade of work by Blanchard and other groups to define the conformational dynamics (and predict intermediates) during translocation.

The major results were obtained by comparing early intermediates in translocation that occur upon EF-G-GTP binding but before GTP hydrolysis, by using the drug spectinomycin that binds to the small subunit and blocks conformational changes required for full translocation. Single-molecule experiments determined conditions with drug that prolonged the early intermediate states (later called INT1) in the presence of EF-G. The authors also solved a later translocation complex with stably bound EF-G-GDP in the presence. A nice feature of this study is that all structures were on *E. coli* ribosomes, by the same group, all at really high cryoEM resolutions ($<2.5\text{\AA}$, except INT2 at lower resolution), allowing atomic-level modeling for many of the conclusions within the manuscript. The first exciting result is the determination of non-EF-G-bound hybrid state structures, first proposed by single-molecule experiments during the past decades. These structures confirmed the existence of these states with distinct shifts of 3' and codon-anticodon positions. Next, EF-G plus spectinomycin is added, and here the fun really starts. The authors observe domain IV of EF-G in contact with the codon-anticodon helix, releasing the standard A-site interactions of the small subunit (A1492, 1493 and G530) with further shift of the tRNA codon-anticodon helices. Thus, is observed a clear molecular basis for EF-G-driven unlocking of the codon-anticodon duplex from the ribosome, and EF-G is unambiguously in an GTP or GDP-Pi state based on the density. Such unlocking is consistent with the fact that non-hydrolyzable GTPs can drive translocation, albeit 50-fold or so more slowly. Also fascinating is the observation that EF-G domain II forms an extended beta-barrel fold in this state, facilitating additional ribosomal contacts that are stabilized in the fully-rotated small subunit state (suggesting that this full rotation state is "activated" for subsequent GTP hydrolysis).

The subsequent state in the authors' ordering is INT2, which is stabilized and populated by the presence of the EF-G inhibitor, fusidic acid, in which GTP is hydrolyzed but subsequent rearrangements of ribosome and factor are slowed (as shown by single-molecule analyses). Here, density for the gamma phosphate on EF-G is lost, EF-G remodels, and the intersubunit

conformation begins a reverse rotation towards its post state. EF-G domain 4 is disengaged from the A-site codon-anticodon complex that has further shifted towards its translocated position in the P site, with relocking of interactions at the A site. The combined structures show a series of discrete shifts of the tRNA-mRNA complexes through the ribosome that are just spectacular to observe (see Fig. 5!). This is a wonderful, carefully performed mix of structural biology guided by dynamics, and reveals key structural aspects of the translocation process, which are likely conserved amongst all organisms. As such, the manuscript is absolutely deserving of publication in Nature, but first needs editing, revision and clarification in a revised version.

1. I found the nomenclatures confusing to follow on first reading (it took me several times through to understand completely, and I am in the field). Perhaps the authors can begin with an expanded Fig. 1 that blows up the schematic (which is tiny) that defines the different states. In particular, the nuances of the different states (rotational state, liganding, positions of the tRNAs) are lost at this small size. Also, the dissociation of the P-site tRNA is not really needed here, allowing simplification of the figure. A structure figure highlighting the positions of the labels would be useful either here or in the supplemental data (there are only schematics).

2. The linkage of the single-molecule data and cryoEM is not made clearly. Essentially, the single-molecule data on inhibited translocation allowed determination of freezing conditions for the given structures. This is fine, but should be explained more clearly in the text. When I first read the manuscript, I thought that they would be doing rapid mixing and real-time freezing. Trapping and stabilizing the intermediates with drugs is fine, but this should be explicitly discussed in the text.

3. More importantly, there is the explicit assumption that drugs do not perturb the mechanisms and pathways. The best support of this is the transient existence of some of the proposed states during uninhibited translocation in the single-molecule experiments. The authors should discuss how the drugs might perturb their mechanistic insights. Clearly, they change the energy landscape of translocation, and this should be stated correctly.

4. Also, EF-G only binds transiently (less than a second) to most of these states (except the FA complexes), as shown by single-molecule studies. Thus, the cryoEM complexes likely represent rebinding of EF-G to the different states after sampling. The authors should note this explicitly, as I do not think it takes away from their conclusions and is a fairer depiction of their structural studies.

5. Some figures and movies could be improved. I find Fig. 5 essential, and the top portion is clear. The bottom panel b showing the codon-anticodon positions is confusing on first glance. These are critical panels, so maybe another effort to present the codon-anticodon positions more clearly is needed. Maybe use thick ribbons for the starting and ending states, and lines or faded colors for the intermediates? As it stands, it looks a bit like a plate of pasta at a fancy restaurant. I really liked the boxed presentation of the E-site tRNAs in ED Fig. 4a. Very clear.

6. The authors could be a bit more generous in their referencing, for example including recent measurements by Rodnina on translocation dynamics, without denigrating their own efforts over the years.

7. I would like to see a rigorous and detailed comparisons of the predictions of single-molecule states (structures, populations) and the cryoEM structures. There are only oblique or non-detailed references, requiring the reader to go back to prior papers. Perhaps a supplemental figure would help here.

8. The supplemental movies with just densities are hard to follow. I would add real molecular movies with ribbon diagrams for the tRNA movements and EF-G changes, while density is fine for the ribosomal domain movements. I would also like to see a movie for the different tRNA conformations (ED Fig. 6).

9. There are several instances where the writing could be toned down. The peristaltic relay in the discussion (not sure what that means), as well as terms such as leading and lagging edge (define explicitly, for non-ribosomologists). In discussion of the INT2 structure there is the statement "We interpret this distinction to represent mesoscopic heterogeneities within the dynamic INT2 basin that reflect the diffusive....", which also sounds overblown.

10. There is a typo on the EF-G bimolecular association rate written μM instead of μM .

These are all relatively minor points, but I hope will help the authors present this work more clearly to the community. This was the most exciting ribosomal structural work I have seen in years, so deserves outstanding presentation.

Referee #2:

The novelty of the study comes from the identification of the so-called INT1 structure, a previously not seen first intermediate. The portion of the paper describing these results goes on as follows:

"While we cannot unambiguously determine if the nucleotide binding pocket of EF-G is bound to GDP and inorganic phosphate (Pi) or GTP, or a mixture of the two in dynamic exchange³⁸, we can conclude that EF-G is capable of unlocking the peptidyl-tRNA cargo from the SSU decoding center prior to Pi release. While previous efforts have trapped pre-hydrolysis EF-G conformations on substrates lacking peptidyl-tRNA cargo or on POST complexes using non-hydrolyzable GTP analogues^{11,12,39} or a catalytically-dead EF-G mutant¹⁷, the early translocation intermediate captured here represents the first example of EF-G engaged with its physiological substrate in a pre-hydrolysis EF-G conformation."

Since it cannot be clarified unambiguously whether the complex presents GTP or GDP.Pi, its characterization as "pre-hydrolysis" -- which is brought out as main news in the title -- cannot be sustained. It is of course understood that in many processes following GTP hydrolysis, the release and use of the liberated energy is delayed until the release of the inorganic phosphate, but this doesn't justify the characterization, by induction, of the complex at hand as "pre-hydrolysis", nor has it been experimentally ascertained in this study that there has been no conformational change between the actual GTP hydrolysis and the conformation observed here.

Author Rebuttals to Initial Comments:

Point-by-Point responses to Referee comments

We wish to thank both Referees for their thoughtful comments and helpful suggestions. Please find our responses to the points raised below. Changes to our manuscript that have been made to address Referee comments are highlighted in blue in our revised manuscript.

Referee #1:

1. I found the nomenclatures confusing to follow on first reading (it took me several times through to understand completely, and I am in the field). Perhaps the authors can begin with an expanded Fig. 1 that blows up the schematic (which is tiny) that defines the different states. In particular, the nuances of the different states (rotational state, liganding, positions of the tRNAs) are lost at this small size. Also, the dissociation of the P-site tRNA is not really needed here, allowing simplification of the figure. A structure figure highlighting the positions of the labels would be useful either here or in the supplemental data (there are only schematics).

RESPONSE: We thank this Referee for these very helpful suggestions. To remedy this, we have:

- (I) Enlarged the schematic in Fig. 1a, added tRNA position labels, and labels to highlight the sites of labeling for Fig. 1b;

Fig. 1 | Early kinetic and structural intermediate of tRNA₂-mRNA translocation. **a**, Schematic of the translocation reaction coordinate in bacteria depicting small subunit (SSU; 30S) rotation (blue) and head swivel (purple) processes. tRNAs are colored on a gradient from the A (green) to P (yellow) to E (orange) sites. Boxed ribosome complexes were characterized in this study. Green (donor-LD555) and red (acceptor-LD650) circles denote positions of smFRET dyes on uS13 and uL1, respectively, used in panel b. **b**, Population FRET histograms showing time evolution of FRET between uS13 (LD555) and uL1 (LD650) upon injection of EF-G with either SPC (3 mM) or FA (400 μ M). N indicates the number of observed molecules, 400 ms time resolution. **c**, Overview of the INT1 ribosome structure captured by SPC

- (II) Added a schematic panel to Extended Data Fig. 2 to assist the non-specialist. The deacyl-tRNA dissociation depicted in Fig. 1a was intended to illustrate that dissociation can occur at different stages, but we understand that it adds unnecessary complexity, so we have removed it from the main text figure (though it is included in the Extended Data Fig. 2 with a more generous explanation).

Extended Data Fig. 2 | Global conformational changes within the ribosome that define the translocation reaction coordinate. a. Schematic of the translocation reaction coordinate in bacteria depicting small subunit (SSU; 30S) rotation (blue) and head swivel (purple) processes. tRNAs are colored on a gradient from the A (green) to P (yellow) to E (orange) sites. Deacyl-tRNA dissociation (orange) can occur at multiple steps after INT2. tRNA positions depicted in chimeric-hybrid notation (ssu head SSU BODY/LARGE SUBUNIT (LSU)). **b.** SSU conformational changes accompanying each sequential translocation step, viewed from inside the intersubunit space (left) and towards the intersubunit space from the head domain (inset), colored by RMSD at each SSU residue for each transition. Degree of shoulder domain closure, SSU body rotation, and SSU head swivel as compared to POST indicated as “total”. Degree changes indicate the axes of SSU body rotation (transparent black, LSU core alignment) and SSU head swivel (solid black, SSU body alignment). **c.** Deacyl-tRNA (yellow/orange), peptidyl-tRNA (green/yellow) and EF-G (red) movements during translocation. Current tRNA and EF-G positioning (solid colored, outlined), previous position (transparent color, no outline), and next position (white, solid outline). Alignment on LSU core. Camera perspective identical for all images. See also Extended Data Table 1.

2. *The linkage of the single-molecule data and cryoEM is not made clearly. Essentially, the single-molecule data on inhibited translocation allowed determination of freezing conditions for the given structures. This is fine, but should be explained more clearly in the text. When I first read the manuscript, I thought that they would be doing rapid mixing and real-time freezing. Trapping and stabilizing the intermediates with drugs is fine, but this should be explicitly discussed in the text.*

RESPONSE: We thank the Referee for bringing the need for additional clarity to the connection between the smFRET and cryo-EM data to our attention. This study would not have been possible without the establishment of pre-steady state smFRET to optimize reaction conditions and timing of our grid preparations. We, indeed, froze pre-steady state translocation reactions performed analogously to those performed in smFRET studies to capture the intermediates identified. We have thus added clarifying language in the first results section to explain more explicitly how the pre-steady state reactions were initiated immediately prior to cryo-EM grid preparation.

3. *More importantly, there is the explicit assumption that drugs do not perturb the mechanisms and pathways. The best support of this is the transient existence of some of the proposed states during uninhibited translocation in the single-molecule experiments. The authors should discuss how the drugs might perturb their mechanistic insights. Clearly, they change the energy landscape of translocation, and this should be stated correctly.*

RESPONSE: We thank the Referee for highlighting the need for greater clarity in this regard. We have worked to make our use of spectinomycin for INT1 capture and FA for INT2 capture more explicitly stated. We have also added language to articulate that the antibiotic-stabilized states are approximations of intermediate states that are rapidly transited in the absence of drug.

4. *Also, EF-G only binds transiently (less than a second) to most of these states (except the FA complexes), as shown by single-molecule studies. Thus, the cryoEM complexes likely represent rebinding of EF-G to the different states after sampling. The authors should note this explicitly, as I do not think it takes away from their conclusions and is a fairer depiction of their structural studies.*

RESPONSE: As the Referee notes, it is formally possible that the earliest translocation intermediate may represent a single EF-G binding event that was stalled for a prolonged time, or it may represent a quasi-equilibrium of ribosomes that are repetitively and reversibly engaged by EF-G. We have updated the manuscript text to more explicitly express this possibility.

5. *Some figures and movies could be improved. I find Fig. 5 essential, and the top portion is clear. The bottom panel b showing the codon-anticodon positions is confusing on first glance. These are critical panels, so maybe another effort to present the codon-anticodon positions more clearly is needed. Maybe use thick ribbons for the starting and ending states, and lines or faded colors for the intermediates? As it stands, it looks a bit like a plate of pasta at a fancy restaurant. I really liked the boxed presentation of the E-site tRNAs in ED Fig. 4a. Very clear.*

RESPONSE: We again thank this Referee for this helpful suggestion. We agree that Fig. 5 clarity is vital in summarizing our findings. We have made the following changes to Fig. 5:

- (I) Separated panel **b** into deacyl- and peptidyl-tRNA panels to make the figure less cluttered and to more clearly demarcate the codon-anticodon pairs.
- (II) Highlighted the movement of tRNA position 34 and mRNA positions +3 (deacyl) and +6 (peptidyl) in panel **b** to make a clear connection between panels **b** and **c**.
- (III) Added reference points to panel **c**, such as “30S head”, “30S body”, “A, P, E sites” to illustrate that the tRNA-mRNA pair moves towards the 30S head domain during intermediate stages of translocation.
- (IV) Added panel **d** to illustrate the stepwise tRNA-mRNA movement, as in Extended Data Fig. 4a.

Fig. 5 | Non-uniform movement of the tRNA₂-mRNA module during translocation. a. Overlay of the tRNA₂-mRNA during each structural intermediate of translocation from the A (green) to P (yellow) to E (orange) sites. Boxed regions depict perspectives shown in panel b. **b.** Overlay from panel a, viewed from the codon-anticodon interface. Circles on the tRNAs at position 34 N1 (deacyl) and N3 (peptidyl) depict the tRNA trajectories during translocation. **c.** Plotted tRNA positions 34 from panel b and distance movement between states. tRNA-mRNA pair moves towards the small subunit (SSU; 30S) head domain (purple) and away from the SSU body domain (blue) in intermediate states of translocation. **d.** tRNA anticodon-mRNA codon movement during translocation, same perspective as panels b, c. See also Extended Data Table 2.

6. The authors could be a bit more generous in their referencing, for example including recent measurements by Rodnina on translocation dynamics, without denigrating their own efforts over the years.

RESPONSE: We sincerely apologize for unintentional referencing omissions. We have carefully revised all citations for completeness to the best of our ability within the limits of the Nature format restrictions.

7. I would like to see a rigorous and detailed comparisons of the predictions of single-molecule states (structures, populations) and the cryoEM structures. There are only oblique or non-detailed references, requiring the reader to go back to prior papers. Perhaps a supplemental figure would help here.

RESPONSE: We thank the Referee for this suggestion and have added Supplementary Fig. 13 and Supplementary Table 2 to assist experts and nonexperts in digesting the connection between single-molecule states and the cryo-EM structures presented here.

Supplementary Fig. 13 | Published FRET pair positions mapped on the INT1 structure. Approximate donor (green sphere) and acceptor (red sphere) fluorescent dye positions used in published FRET studies. **a**, donor tRNA^{Phe} (Cy3-s⁴U8) in the P site to acceptor fMet-Phe-Lys-tRNA^{Lys} (Cy5-acp³U47) in the A site from Munro et al, 2007^{5,6}, **b**, donor tRNA^{Phe} (Cy3-acp³U47) in the P site to acceptor fMet-Phe-Lys-tRNA^{Lys} (Cy5-acp³U47) in the A site from Wasserman et al, 2016⁶, **c**, donor uS13 (LD550- N terminal ACP) to acceptor fMet-Phe-Lys-tRNA^{Lys} (Cy5-acp³U47) in the A site from Wasserman et al, 2016⁶, **d**, donor uL11 (Cy3-residue 87) to acceptor fMet-Phe-Lys-tRNA^{Lys} (Cy5-acp³U47) in the A site from Chen et al, 2011⁷, **e**, donor uS13 (LD550- N terminal ACP) to acceptor uL5 (LD650- N terminal ACP) from Wasserman et al, 2016⁶, **f**, donor bL9 (Cy3-N11C) to acceptor bS6 (Cy5-D41C) from Cornish et al, 2008⁸, **g**, donor fMet-Phe-Lys-tRNA^{Lys} (Cy3-acp³U47) in the A site to acceptor EF-G (Cy5-C terminal SFP) from Munro et al, 2011⁹, **h**, donor EF-G (bifunctional rhodamine 467-474) to acceptor uL11 (Cy5-residue 87) from Chen et al, 2016¹⁰. Camera perspective identical for all images. See also Supplementary Table 2.

Supplementary Table 2 | Published FRET-pair positions, estimated FRET efficiencies reported and approximate distances from the points of fluorophore attachment in PRE-C, PRE-H2*, PRE-H1, INT1, INT2, and POST.

Signal	Donor		Acceptor		Publication	FRET Efficiency						
	PRE-C	PRE-H2*	PRE-H1	INT1		INT2	POST					
tRNA-tRNA	tRNA ^{fMet}	Cy3-s ⁴ U8	fMet-Phe-tRNA ^{Phe}	Cy5-acp ³ U47	Munro et al, 2007	~0.55	~0.24	~0.39	-	-	-	
	tRNA ^{Phe}	Cy3-s ⁴ U8	fMet-Phe-Lys-tRNA ^{Lys}	Cy5-acp ³ U47	Wasserman et al, 2016	0.61		0.33	~0.37	~0.6	0.61	
						DISTANCE (Å)	~47	~65	~62	~57	~53	~54
tRNA-tRNA	tRNA ^{Arg}	Cy3-acp ³ U47	fMet-Arg-Phe-tRNA ^{Phe}	Cy5-acp ³ U47	Chen et al, 2011	0.69		0.38	-	-	0.55	
							DISTANCE (Å)	~38	~65	~56	~52	~45
S13-A site	uS13	LD550-ACP-N	fMet-Phe-Lys-tRNA ^{Lys}	Cy5-acp ³ U47	Wasserman et al, 2016		0.13			~0.15	0.34	
							DISTANCE (Å)	~89	~97	~87	~89	~90
								~91*		~89.5*		
uL11-A site	uL11	Cy3-87	fMet-Arg-Phe-tRNA ^{Phe}	Cy5-acp ³ U47	Chen et al, 2011	0.62		0.34	-	-	0.15	
							DISTANCE (Å)	~46	~53	~53	~60	~60
uS13-uL5	uS13	LD550-ACP-N	uL5	LD650-ACP-N	Wasserman et al, 2016	0.76		0.56	~0.5	~0.45	0.76	
							DISTANCE (Å)	32	~46	~47	~51	~59
uL9-uS6	bL9	Cy3-(N11C)	bS6	Cy5-(D41C)	Cornish et al, 2008	0.56		-0.4	-	-	-	
							DISTANCE (Å)	~55	~67	~67	~66	~64
A site-EF-G	tRNA ^{Phe}	Cy3-acp ³ U47	EF-G	C-SFP-Cy5	Munro et al, 2011	-	-	-	~0.73	~0.55	~0.55	
							DISTANCE (Å)	-	-	-	~24	~29

Distances are approximate and do not explicitly account for experimental uncertainties in the dye positions and FRET values. *Values are averaged. See also Supplementary Fig. 13. Distance measurements are made between the points of fluorophore attachment.

8. *The supplemental movies with just densities are hard to follow. I would add real molecular movies with ribbon diagrams for the tRNA movements and EF-G changes, while density is fine for the ribosomal domain movements. I would also like to see a movie for the different tRNA conformations (ED Fig. 6).*

RESPONSE: We fully agree, and we have made additional effort to clarify the presentation of information in the Supplemental Videos:

- (I) **Supplementary Video 1** | tRNA movement through the intersubunit space during translocation. This video features ribbon diagrams and cryo-EM density to depict tRNA movement through the six states of tRNA/mRNA translocation.
- (II) **Supplementary Video 5** | tRNA conformational changes during translocation. This video depicts the conformational heterogeneity of deacyl- and peptidyl-tRNA during translocation, as in Extended Data Fig. 6.
- (III) **Supplementary Video 7** | Conformational change in EF-G during translocation. This video illustrates the observed conformational change in EF-G during the INT1 to INT2 transition from multiple perspectives.

9. *There are several instances where the writing could be toned down. The peristaltic relay in the discussion (not sure what that means), as well as terms such as leading and lagging edge (define explicitly, for non-ribosomologists). In discussion of the INT2 structure there is the statement “We interpret this distinction to represent mesoscopic heterogeneities within the dynamic INT2 basin that reflect the diffusive...”, which also sounds overblown.*

RESPONSE: We thank the Referee for bringing this to our attention. In writing this manuscript, we endeavored to find terminologies to adequately describe our findings and occasionally used atypical language. We have now revised our text to remove such language and to clearly define the terminologies we utilize. The terms leading and lagging edge are have been employed in the transcription and replication literature and we believe their application to the translating ribosome will help reader comprehension. To aid the reader, we added “3'-end of mRNA” and “5'-end of mRNA” when these terms are first introduced. In addition, we have also made an effort to eliminate instances of verbose or speculative language from the text. In this context, we have reduced the complexity of the specific sentence referenced by the Referee, while retaining the key point that the distinctions observed in the INT2 complex are consistent with previous literature (Wasserman et al. NSMB 2016, in particular) and structurally relevant considerations.

10. *There is a typo on the EF-G bimolecular association rate written μm instead of μM .*

RESPONSE: This typo has been fixed. We are grateful for this Referee’s careful and thorough review of the manuscript.

Referee #2:

Since it cannot be clarified unambiguously whether the complex presents GTP or GDP.Pi, its characterization as “pre-hydrolysis” -- which is brought out as main news in the title -- cannot be sustained. It is of course understood that in many processes following GTP hydrolysis, the release and use of the liberated energy is delayed until the release of the inorganic phosphate, but this doesn’t justify the characterization, by induction, of the complex at hand as “pre-hydrolysis”, nor has it been experimentally ascertained in this study that there has been no conformational change between the actual GTP hydrolysis and the conformation observed here.

RESPONSE: We thank the Referee for bringing this point of potential confusion and miscommunication to our attention. We have now revised the title to remove terminology related to GTP hydrolysis and to more broadly encompass the full scope of the manuscript. This comment inspired a more rigorous comparative analysis of our structure and the terminologies used to describe GTPase activation and catalysis. We now follow pervasive literature precedent in this regard by replacing instances of “pre-hydrolysis” conformation with “active, GTP” conformation. We hope that these analyses and revisions instill additional confidence in our conclusions and adequately address this concern.

In the writing of this manuscript, we took strides to clarify that we have captured EF-G in a conformation that closely resembles what is expected for the GTP-bound state, immediately before or after GTP hydrolysis. In retrospect, we realize that readers could be confused by our reference to a “pre-hydrolysis conformation” as the complex could represent a state immediately after GTP hydrolysis that closely resembles the “pre-hydrolysis” EF-G conformation.

To further clarify this point, we have performed an in-depth analysis of prior high-resolution structural studies of other soluble G proteins bound to GTP, non-hydrolysable GTP analogues, and GDP-P_i to substantiate that the G domain elements in EF-

G are poised very similarly to an active, GTP-bound conformation. These comparisons are now reported in Extended Data Fig. 11 and Supplemental Figs. 14-16 and are summarized concisely below.

Elements assigned to the active, GTP conformation of EF-G.

Switch I: A structured switch I element in Ras-like GTPases (also known as G2) is canonically associated with an active, GTP-bound state in which a conserved threonine (Thr61 in EF-G) coordinates with a Mg^{2+} ion that bridges the β and γ phosphates of the bound nucleotide¹. The switch I element is also responsible for providing the binding surface for effector molecules in classical signaling pathways². In our INT1 structure, switch I encircles the β and γ phosphates of the bound nucleotide and engages in both inter and intra molecular contacts that support an active state. These include C-terminal amphipathic helical interaction with domain III and domain II binding, which will be discussed in detail in the following sections.

Switch II: Structuring of the switch II loop (DXXGQ/H/T, also known as G3) helps coordinate the γ phosphate in the active GTP-bound conformation¹. The residue associated with facilitating the water mediated nucleophilic attack of the γ phosphate (His92 in EF-G) is located on the switch II loop. When bound to GTP, this residue is positioned in close proximity to the terminal γ -phosphate. In our INT1 structure, switch II is ordered and directly engaged with the γ -phosphate via His92, supporting an active state.

Domain II: In the majority of EF-G structures, domain II adopts a classic β -barrel fold (Extended Data Fig. 11). However, in our INT1 structure, the 369 loop of domain II rearranges to contact switch I. This conformational change has only been observed in GTP-bound structures of EF-G, as discussed below. We also note that the switch I to domain II interaction is reminiscent of Ras-effector complex structures, providing additional support that EF-G in INT1 is in an active state (additional discussion below).

INT1 G-domain features supporting an active GTP-bound conformation.

Utilizing structural and biochemical studies of Ras family GTPases, we sought to identify features of EF-G in INT1 that define it as adopting an active, GTP-bound conformation. Crystallographic studies of H-Ras define switch I Thr35 (G2) and switch II Gly60/Gln61 (G3) as “spring loaded” elements that coordinate the Mg^{2+} ion and the terminal phosphate — adapting to the nucleotide state of the G domain and controlling the active/inactive transition². Through an analysis of H-Ras in GDP-, GTP-, and GDPNP-bound states, we found that the catalytic elements of switch II and the coordinating switch I threonine in our INT1 structure align with the activated GTP/GDPNP-states of H-Ras (Fig. R1). By contrast, in the GDP-bound structure of H-Ras, switch I relaxes away from the nucleotide binding pocket with Thr35 flipped away from the Mg^{2+} .

We have added a Supplementary Fig. 14 documenting the comparison between EF-G and Ras. **Similarities between EF-G G domain elements and published EF-Tu(GTP) aa-tRNA structures.** Core GTPase elements of EF-G in our INT1 structure closely resemble extant structures of EF-Tu in its active, GTP-bound form (Fig. R2). This includes, but is not limited to, ordering of switch I C terminus, Mg^{2+} coordination by Thr62, hydrophobic gate positioning

Fig. R1 | Structural comparison of EF-G in INT1 and H-Ras(GTP). Overlay of GTP/GDP-P_i conformation of EF-G (INT1; colored by domain) and H-Ras (gray, PDB-ID 1qra)¹¹.

Fig. R2 | Structural comparison of EF-G in INT1 and EF-Tu(GTP)aa-tRNA. GTP/GDP-P_i conformation of EF-G (INT1; colored) compared to published structure of EF-Tu (gray, PDB-ID 6wd2¹²) overall (left) and in the nucleotide binding pocket (inset, right). Alignment on the G domain.

(Phe95), and positioning of His92 with respect to the terminal phosphate. We have added a Supplementary Fig. 15 documenting this comparison between EF-G and EF-Tu.

Similarities between EF-G switch I and domain II elements and published EF-G(GTP) structures.

Following a rigorous comparison of all EF-G structures available to the public (summarized in Extended Data Fig. 11), we found strong similarities between elements of the INT1 structure presented in our manuscript and the structure of a thermophilic EF-G homolog EF-G-2 bound to GTP (Fig. R3a; PDB-ID 1wdt)³. This crystal structure not only possesses the same extended switch I architecture, but it also displays the same modified domain II β barrel position. The β -barrel positioning was also observed in two structures of a H92A mutant of EF-G bound GTP on the 70S ribosome (Fig. R3b; PDB-ID 3jai and 3j9z)⁴. This later comparison was not directly cited in our initial submission but has now been added. These structural comparisons support that INT1 reflects an active GTP-bound EF-G conformation. We have added panels to Extended Data Fig. 11 with close-ups of the nucleotide binding pocket for high resolution structures.

Fig. R3 | Structural comparison of EF-G in INT1 and published EF-G(GTP) conformations. GTP/GDP-P_i conformation of EF-G (INT1; colored) compared to published structures (gray) of **a**, EF-G-H92A(GTP) (PDB-ID 3jai)⁴ and **b**, an EF-G homolog EFG-2(GTP) (PDB-ID 1wdt)³. Conformational change in domain II indicated with an arrow. Alignment on the G domain.

Overlapping binding sites of EF-G switch I and Ras-Effector complexes. Ras is considered to be a classic molecular switch that binds to GTPase-activating proteins (GAPs) and effector molecules while in its active GTP-bound conformation to drive downstream signaling processes. These binding partners conventionally bind Ras via its switch I loop. Alignment of Ras-complex structures with the EF-G conformation observed in our INT1 structure revealed that the modified domain II fold sits in the same position as classic Ras effector molecules (Fig. R4). As effector molecules are engaged in active conformations, this supports that EF-G is in an active conformation in our INT1 structure. We have added a Supplementary Fig. 16 documenting this comparison.

Fig. R4 | Structural comparison of EF-G in INT1 and H-Ras complexes. **a-c**, Overlays of GTP/GDP-P_i conformation of EF-G (INT1; colored by domain) and H-Ras (light gray) in complex with binding partners. **a**, H-Ras(GDPNP)-Raf(RBD) (PDB-ID 4g0n¹³); **b**, H-Ras(GTP)-PLC- ϵ (PDB-ID 2c5¹⁴); **c**, H-Ras(GDPNP)-NORE1A (PDB-ID 3ddc¹⁵). Arrow designates to EF-G switch I to domain II interaction. Alignment on the G domain.

References for Response to Referees:

1. Vetter, I. R. & Wittinghofer, A. The guanine nucleotide-binding switch in three dimensions. *Science* **294**, 1299–1304 (2001).
2. Gasper, R. & Wittinghofer, F. The Ras switch in structural and historical perspective. *Biol. Chem.* **401**, 143–163 (2019).
3. Connell, S. R. *et al.* Structural basis for interaction of the ribosome with the switch regions of GTP-bound elongation factors. *Mol. Cell* **25**, 751–764 (2007).
4. Li, W. *et al.* Activation of GTP hydrolysis in mRNA-tRNA translocation by elongation factor G. *Sci. Adv.* **1**, (2015).
5. Munro, J. B., Altman, R. B., O'Connor, N. & Blanchard, S. C. Identification of two distinct hybrid state intermediates on the ribosome. *Mol. Cell* **25**, 505–517 (2007).
6. Wasserman, M. R., Alejo, J. L., Altman, R. B. & Blanchard, S. C. Multiperspective smFRET reveals rate-determining late intermediates of ribosomal translocation. *Nat. Struct. Mol. Biol.* **23**, 333–341 (2016).
7. Chen, C. *et al.* Single-molecule fluorescence measurements of ribosomal translocation dynamics. *Mol. Cell* **42**, 367–377 (2011).
8. Cornish, P. V., Ermolenko, D. N., Noller, H. F. & Ha, T. Spontaneous intersubunit rotation in single ribosomes. *Mol. Cell* **30**, 578–588 (2008).
9. Munro, J. B., Wasserman, M. R., Altman, R. B., Wang, L. & Blanchard, S. C. Correlated conformational events in EF-G and the ribosome regulate translocation. *Nat. Struct. Mol. Biol.* **17**, 1470–1477 (2010).
10. Chen, C. *et al.* Elongation factor G initiates translocation through a power stroke. *Proc. Natl. Acad. Sci. USA* **113**, 7515–7520 (2016).
11. Scheidig, A. J., Burmester, C. & Goody, R. S. The pre-hydrolysis state of p21(ras) in complex with GTP: new insights into the role of water molecules in the GTP hydrolysis reaction of ras-like proteins. *Structure* **7**, 1311–1324 (1999).
12. Loveland, A. B., Demo, G. & Korostelev, A. A. Cryo-EM of elongating ribosome with EF-Tu•GTP elucidates tRNA proofreading. *Nature* **584**, 640–645 (2020).
13. Fetits, S. K. *et al.* Allosteric effects of the oncogenic RasQ61L mutant on Raf-RBD. *Structure* **23**, 505–516 (2015).
14. Bunney, T. D. *et al.* Structural and mechanistic insights into ras association domains of phospholipase C epsilon. *Mol. Cell* **21**, 495–507 (2006).
15. Stieglitz, B. *et al.* Novel type of Ras effector interaction established between tumour suppressor NORE1A and Ras switch II. *EMBO J.* **27**, 1995–2005 (2008).

Reviewer Reports on the First Revision:

Referee #1:

Remarks to the Author:

The authors have carefully and diligently addressed my scientific and presentation concerns in the revised manuscript and figures. Both are far clearer, and do greater justice to this fine work. In the final submission, the authors may want to consider eliminating the use of the outlined text fonts in the figures, as I find them often unclear to follow in the context of the figures. Just one last thought.

That said, this manuscript is now acceptable for publication in Nature.

Jody Puglisi

Author Rebuttals to First Revision:

N/A